# Somatic increase of CCT8 mimics proteostasis of human pluripotent stem cells and extends *C. elegans* lifespan

Alireza Noormohammadi[1,*], Amirabbas Khodakarami[1,*], Ricardo Gutierrez-Garcia[1], Hyun Ju Lee[1], Seda Koyuncu[1], Tim König[1], Christina Schindler[1], Isabel Saez[1], Azra Fatima[1], Christoph Dieterich[2] & David Vilchez[1]

Human embryonic stem cells can replicate indefinitely while maintaining their undifferentiated state and, therefore, are immortal in culture. This capacity may demand avoidance of any imbalance in protein homeostasis (proteostasis) that would otherwise compromise stem cell identity. Here we show that human pluripotent stem cells exhibit enhanced assembly of the TRiC/CCT complex, a chaperonin that facilitates the folding of 10% of the proteome. We find that ectopic expression of a single subunit (CCT8) is sufficient to increase TRiC/CCT assembly. Moreover, increased TRiC/CCT complex is required to avoid aggregation of mutant Huntingtin protein. We further show that increased expression of CCT8 in somatic tissues extends *Caenorhabditis elegans* lifespan in a TRiC/CCT-dependent manner. Ectopic expression of CCT8 also ameliorates the age-associated demise of proteostasis and corrects proteostatic deficiencies in worm models of Huntington's disease. Our results suggest proteostasis is a common principle that links organismal longevity with hESC immortality.

[1] Cologne Excellence Cluster for Cellular Stress Responses in Aging-Associated Diseases (CECAD), University of Cologne, Joseph Stelzmann Strasse 26, Cologne 50931, Germany. [2] Section of Bioinformatics and Systems Cardiology, Department of Internal Medicine III and Klaus Tschira Institute for Computational Cardiology, Neuenheimer Feld 669, University Hospital, Heidelberg 69120, Germany. * These authors contributed equally to this work. Correspondence and requests for materials should be addressed to D.V. (email: dvilchez@uni-koeln.de).

The survival of an organism is linked to its ability to maintain the integrity of the cellular proteome. Oxidative or thermal stress, misfolding-prone mutations and aging challenge the structure of proteins. Damaged proteins form toxic aggregates and disrupt cellular membranes, overwhelming the cellular machinery required for their degradation and causing cell malfunction and death[1]. The quality of the proteome is controlled by proteostasis, a complex network of competing and integrated cellular pathways that regulate the synthesis, folding, aggregation, trafficking, interaction and degradation of proteins[2]. With age, post-mitotic cells lose extensive control of proteostasis: widespread, aberrant changes in translation, a generalized downregulation of chaperones and a loss of function in protein degradation machineries often appear in differentiated cells across time[3,4]. This demise in proteostasis is considered one of the hallmarks of aging and contributes to the functional loss characteristic of old organisms[2,3]. Proteostasis dysfunction is associated with multiple age-related disorders such as Huntington's disease (HD), Alzheimer's or Parkinson's disease[2,3]. In contrast, preservation or enhancement of proteostasis surveillance systems until late in life improves resistance to proteotoxic stress and slows down the aging process[4,5].

Embryonic stem cells (ESCs) demonstrate a striking capacity to proliferate indefinitely while maintaining their pluripotency[6]—a capacity that necessarily demands avoidance of any imbalance in proteostasis that would otherwise compromise their function and immortality. Furthermore, ESCs require high global translational rates to maintain their pluripotency[7]. These observations raise an intriguing question: how do ESCs maintain the quality of their proteome under enhanced protein synthesis and proliferation rates? Thus, we hypothesize that ESCs can provide a novel paradigm to study proteostasis and its demise with age. Human ESCs (hESCs) and induced pluripotent stem cells (iPSCs) exhibit high proteasome activity compared with their differentiated counterparts[8]. However, pluripotent stem cells are remarkably more sensitive ($100 \times$) to proteasome inhibitors than progenitor or terminally differentiated cells[8,9]. Increased proteasome activity of human pluripotent stem cells is induced by enhanced levels of the proteasome subunit PSMD11/RPN-6 (ref. 8). Interestingly, RPN-6 overexpression (OE) in somatic tissues is sufficient to induce proteotoxic resistance and extend healthspan in the organismal model Caenorhabditis elegans[10]. Besides increased proteasome activity, we hypothesize that other proteostasis components are also enhanced in hESCs/iPSCs to maintain their pluripotency and immortality.

In this study, we focused on the chaperome network, a key node of proteostasis. The human chaperome is formed by 332 chaperones and co-chaperones that regulate the folding and function of proteins[11]. The binding of chaperones to nascent proteins assists their folding into the correct structure. Furthermore, chaperones assure the proper folding and cellular localization of proteins throughout their life cycle[12]. Gene expression analysis of human brain aging shows a striking repression of 32% of the chaperome, including ATP-dependent chaperone machines such as cytosolic HSP90, HSP70 family members (for example, HSPA8 and HSPA14) and subunits of the T-complex protein-1 ring complex/chaperonin containing TCP1 (TRiC/CCT) complex. In contrast, 19.5% of the chaperome is induced during human brain aging[11]. In addition, these repression and induction are enhanced in the brains of those with HD, Alzheimer's or Parkinson's disease compared with their age-matched controls[11]. Notably, knockdown of specific chaperome components that are repressed during aging such as CCT subunits, HSPA14 or HSPA8 induces proteotoxicity in HD C. elegans and mammalian cell models[11]. Therefore, defining

differences in the levels and regulation of chaperone machines between immortal hESCs and their differentiated counterparts could be of central importance not only for understanding hESC identity but also the aging process.

Here we show that human pluripotent stem cells exhibit increased assembly of the chaperonin TRiC/CCT complex, a mechanism induced by high levels of specific CCT subunits. By studying proteostasis of pluripotent stem cells, we find that CCT8 is sufficient to increase TRiC/CCT assembly. Furthermore, enhanced TRiC/CCT assembly is required for the striking ability of pluripotent stem cells to maintain proteostasis of aggregation-prone huntingtin (HTT), the mutant protein underlying HD. Our results indicate that the differentiation process correlates with a decline in the expression of CCT subunits and TRiC/CCT assembly. Since the levels of CCT subunits are further decreased in somatic tissues during organismal aging[11], we examined whether modulation of CCT8 can delay the aging process and proteostasis dysfunction by using C. elegans as a model organism. Notably, upregulation of CCT8 levels in somatic tissues triggers TRiC/CCT assembly and extends organismal lifespan particularly under proteotoxic conditions. Thus, we define CCT8 as a key modulator of TRiC/CCT assembly and establish a link between TRiC/CCT chaperonin, hESC identity and youthfulness.

## Results

**CCT subunits decrease during hESC differentiation.** To examine changes in the chaperome network during differentiation, we performed quantitative analysis of both the transcriptome and proteome comparing hESCs with their neural progenitor cell (NPC) and neuronal counterparts (Supplementary Figs 1 and 2 and Supplementary Data 1 and 2). In our transcriptome analysis, we identified 279 chaperome components. Among them, 119 genes were downregulated and 44 genes were upregulated during differentiation into NPCs (Supplementary Fig. 1 and Supplementary Data 1). At the protein level, we found that 36 out of 122 identified chaperome components decrease during differentiation into NPCs (Supplementary Fig. 2 and Supplementary Data 2). In contrast, 27 chaperome components were increased during neural differentiation. Among the 44 chaperome components decreased in terminally differentiated neurons compared with hESCs, 28 proteins were already downregulated during differentiation into NPCs (Supplementary Data 2).

Notably, several subunits of the chaperonin TRiC/CCT complex decreased at both transcript and protein levels during neural differentiation of hESCs (Supplementary Fig. 3 and Supplementary Table 1). TRiC/CCT is required for cell viability as a key component of the proteostasis network that facilitates the folding of $\sim 10\%$ of the eukaryotic proteome[13,14]. TRiC/CCT not only assists the folding of newly synthesized proteins[15] but also binds to misfolded proteins regulating their aggregation[16]. In eukaryotes, the hetero-oligomeric TRiC/CCT complex consists of two stacked rings of eight paralogous subunits each[17]. In our quantitative proteomics assay, we found that CCT3, CCT4 and CCT8 subunits are significantly increased in hESC compared with NPCs (Supplementary Table 1). We confirmed these results by western blot analysis and found that other subunits also decrease during neural and neuronal differentiation (that is, CCT2, CCT6A and CCT7; Fig. 1a). The decrease in the protein amount of CCT subunits correlated with a reduction of the mRNA levels during differentiation (Fig. 1b). This downregulation of CCT subunits was not a specific phenomenon associated with the neural lineage as differentiation into either endoderm or mesoderm induced a similar decrease (Fig. 1c–f). Because hESC lines can vary in their characteristics, we differentiated a distinct hESC line as well as two iPSC lines,

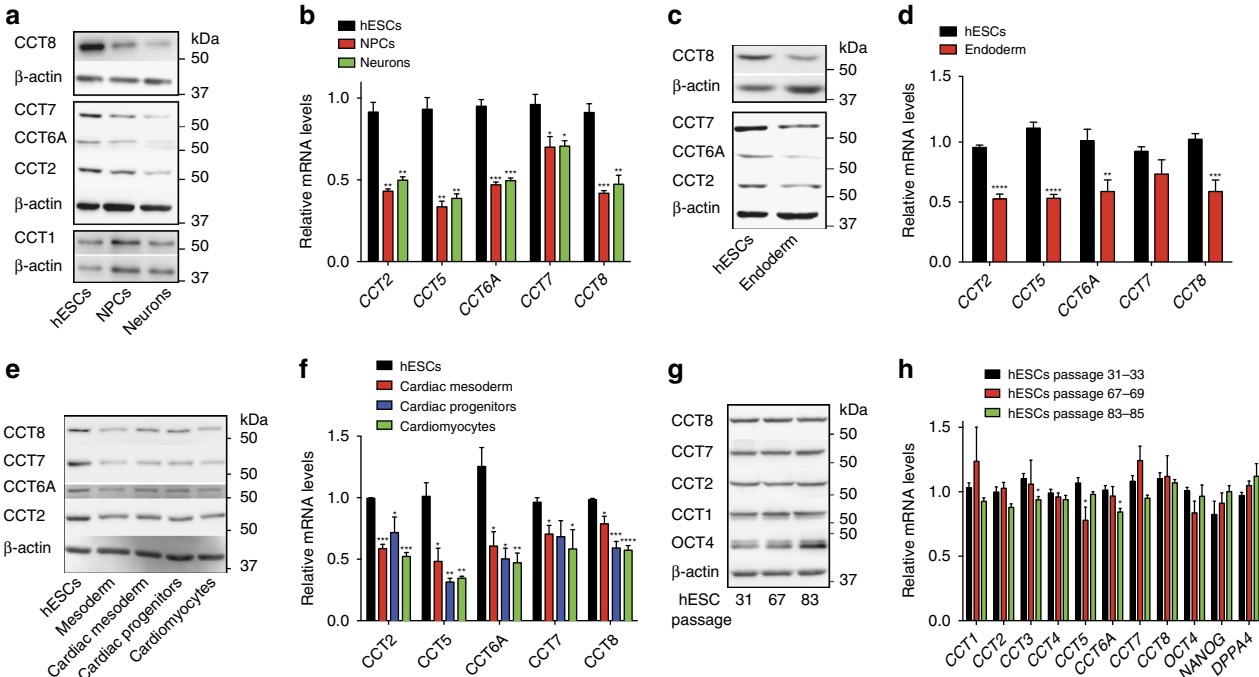

**Figure 1 | The expression of CCT subunits decreases during differentiation.** (**a**,**c**,**e**) Western blot analysis with antibodies to CCT8, CCT7, CCT6A, CCT2 and CCT1. β-actin is the loading control. The images are representative of two independent experiments. In **a**,**c** the differentiation was performed with the H9 hESC line whereas in **e** we used H1 hESCs. (**b**) CCT subunits relative expression to H9 hESCs represents the mean ± s.e.m. of three independent experiments. (**d**) CCT subunits relative expression to H9 hESCs represents the mean ± s.e.m. ($n = 7$ independent experiments). (**f**) CCT subunits relative expression to H1 hESCs represents the mean ± s.e.m. of three independent experiments. (**g**) The protein levels of CCT subunits do not differ with passage in H9 hESCs. β-actin is the loading control. The images are representative of two independent experiments. (**h**) Data represent the mean ± s.e.m. of the relative expression levels to H9 hESCs passage 31–33 ($n = 3$ independent experiments). All the statistical comparisons were made by Student's $t$ test for unpaired samples. $P$ value: $^*P < 0.05$, $^{**}P < 0.01$, $^{***}P < 0.001$, $^{****}P < 0.0001$.

and obtained similar results (Supplementary Figs 4 and 5). Consistent with the ability of pluripotent stem cells to self-renew indefinitely while maintaining their undifferentiated state[6], the expression of CCT subunits and pluripotency markers did not decline with passage (Fig. 1g,h). Taken together, our results indicate that human pluripotent stem cells are able to maintain enhanced expression of CCT subunits under unlimited proliferation in their undifferentiated state. However, the levels of subunits such as CCT8 or CCT2 decrease when hESCs/iPSCs differentiate into distinct cell lineages. Thus, increased levels of CCT subunits could be an intrinsic characteristic of human pluripotent stem cells linked to their immortality and identity.

**Increased expression of CCT8 induces TRiC/CCT assembly.** Prompted by these findings, we asked whether increased levels of CCT subunits resulted into more assembled TRiC/CCT complexes in human pluripotent stem cells. Although both hESCs and NPCs have similar levels of total CCT1 subunits (Figs 1a and 2a, Supplementary Fig. 4 and Supplementary Table 1), hESCs lines exhibited a dramatic increase in the assembly of TRiC/CCT in the form of two stacked rings (Fig. 2a and Supplementary Fig. 4). Similarly, iPSCs also had increased TRiC/CCT assembly compared with differentiated cells (Supplementary Fig. 5). Since all the subunits are required for TRiC/CCT function[18], down-regulated CCT subunits could become structural limiting factors during differentiation and modulate the decrease of TRiC/CCT assembly. An intriguing possibility is that specific subunits can also function as assembly activators. A comparison between the levels (relative to CCT1) of the different subunits in hESCs and NPCs showed that CCT8 is the most abundant subunit in both

cell types despite decreasing during differentiation (Fig. 2b). These findings indicate that CCT8 is not stoichiometric limiting for TRiC/CCT assembly. Thus, we asked whether an increase in the total protein levels of CCT8 could trigger TRiC/CCT assembly. Strikingly, ectopic expression of CCT8 induced an increase in TRiC/CCT assembly whereas the total protein levels of CCT1 remained similar (Fig. 2c). In contrast, we found a decrease in the levels of CCT1 in its monomeric form on CCT8 OE (Fig. 2c). Collectively, our results suggest that human pluripotent stem cells have an intrinsic proteostasis network characterized by high levels of TRiC/CCT complex. However, the levels of several CCT subunits decrease during differentiation resulting in diminished assembly of TRiC/CCT chaperonin. In addition, we identified CCT8 as a potential activator of TRiC/CCT assembly by using hESCs/iPSCs as a model to study proteostasis.

**Loss of CCT subunits affects hESC identity.** With the strong connection between hESC/iPSC identity, CCT8 expression and enhanced assembly of TRiC/CCT complex, we asked whether increased levels of CCT8 contribute to maintain pluripotency. Given that TRiC/CCT is essential for cell viability[13], loss of CCT subunits induced cell death and detachment of hESCs (Supplementary Fig. 6a). To avoid these effects, we induced a mild knockdown of ∼30–50% (Fig. 3a). Notably, downregulation of CCT8 resulted in decreased levels of pluripotency markers (Fig. 3a). We hypothesized that high CCT8 expression impinges on hESC function via enhanced assembly of TRiC/CCT complex. Since mutation or loss of a single subunit is sufficient to impair the activity of the complex[18], we knocked down other subunits

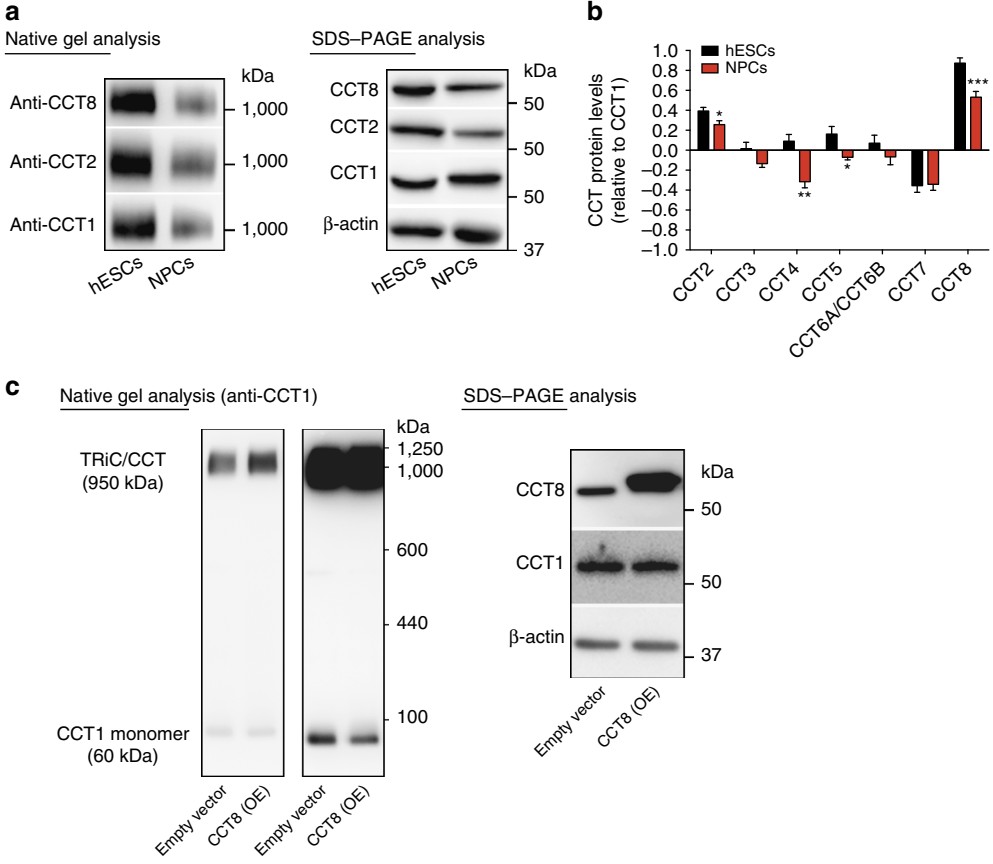

**Figure 2 | Ectopic expression of CCT8 is sufficient to increase TRiC/CCT assembly.** (**a**) Native gel electrophoresis of H9 hESCs and NPCs extracts followed by immunoblotting with CCT antibodies. Extracts were resolved by SDS–PAGE and immunoblotting for analysis of total CCT subunit levels and β-actin loading control. The images are representative of three independent experiments. (**b**) Label-free quantification (LFQ) of CCT protein levels relative to CCT1. All detected CCT subunits were quantified by their $\log_2$ fold change in LFQ intensities relative to CCT1. Graphs represent the mean ± s.e.m. (hESCs ($n = 9$), NPCs ($n = 6$)). All the statistical comparisons were made by Student's t-test for unpaired samples. P-value: *$P < 0.05$, **$P < 0.01$, ***$P < 0.001$. (**c**) Native gel electrophoresis of HEK293T cell extracts followed by immunoblotting with CCT1 antibody (two different exposure times of the same membrane are shown). Extracts were resolved by SDS–PAGE and immunoblotting for analysis of total CCT8 and CCT1 subunit levels. β-actin is the loading control. The images are representative of three independent experiments.

(that is, *CCT2*, *CCT6A* and *CCT7*) to determine whether increased TRiC/CCT is required for hESC function. As with CCT8 knockdown, decreased levels of other subunits affected the expression of pluripotency markers (Fig. 3a,b). We performed these experiments in an independent hESC line as well as two iPSC lines and obtained similar results (Supplementary Fig. 6b–d). In addition, loss of CCT levels induced the expression of markers of the distinct germ layers (Fig. 3c and Supplementary Fig. 7). Since we observed an upregulation in specific markers of the three germ layers (Fig. 3c and Supplementary Fig. 7), our data suggest that hESCs/iPSCs undergo a decline of pluripotency on knockdown of CCT subunits but they do not differentiate into a particular cell lineage. Although we cannot discard a role of free monomeric CCT subunits, our results indicate that increased levels of the TRiC/CCT complex are required for human pluripotent stem cell identity.

**TRiC determines proteostasis of pluripotent stem cells.** Because the TRiC/CCT complex modulates aggregation of damaged and misfolded proteins[16], we asked whether this chaperonin is required for increased proteostasis of pluripotent stem cells. To examine this hypothesis, we used iPSCs derived from HD. HD is an autosomal dominant neurodegenerative disorder caused by the expansion of a CAG triplet repeat region in the *huntingtin* gene

(*HTT*), which translates into a polyglutamine stretch (polyQ) in the protein resulting in proteotoxicity and aggregation[19]. In its wild-type form, HTT contains 6–35 glutamine residues. However, in individuals affected by HD, it contains > 35 glutamine residues and longer repeats predict younger disease onset[19]. Here we used three different control iPSC lines and HD-iPSCs from four donors with different allelic series (that is, polyQ57, polyQ60, polyQ71 and polyQ180). These HD-iPSC lines possess one mutant copy of *huntingtin* gene but also one normal allele (Fig. 4a,b). Although the levels of mutant HTT were lower compared with the normal protein, HD-iPSCs exhibited significant amounts of mutant polyQ-expanded HTT protein (Fig. 4a,b). In iPSCs that express longer CAG repeat expansions (polyQ180), the differences between the levels of mutant and normal HTT were more dramatic (Fig. 4b). Nevertheless, we confirmed the expression of mutant HTT in these iPSCs by using an antibody that detects remarkably better polyQ-expanded HTT than wild-type HTT (refs 20,21; Fig. 4b,e). To examine the aggregation of mutant HTT in HD-iPSCs, we performed filter trap analysis that allows for quantification of polyQ aggregates[10]. Although HD-iPSCs expressed significant levels of polyQ-expanded HTT, these cells did not exhibit accumulation of detectable polyQ aggregates compared with control iPSCs as assessed by both filter trap and immunohistochemistry analyses (Fig. 4c,e and Supplementary Fig. 8). In HD organismal models

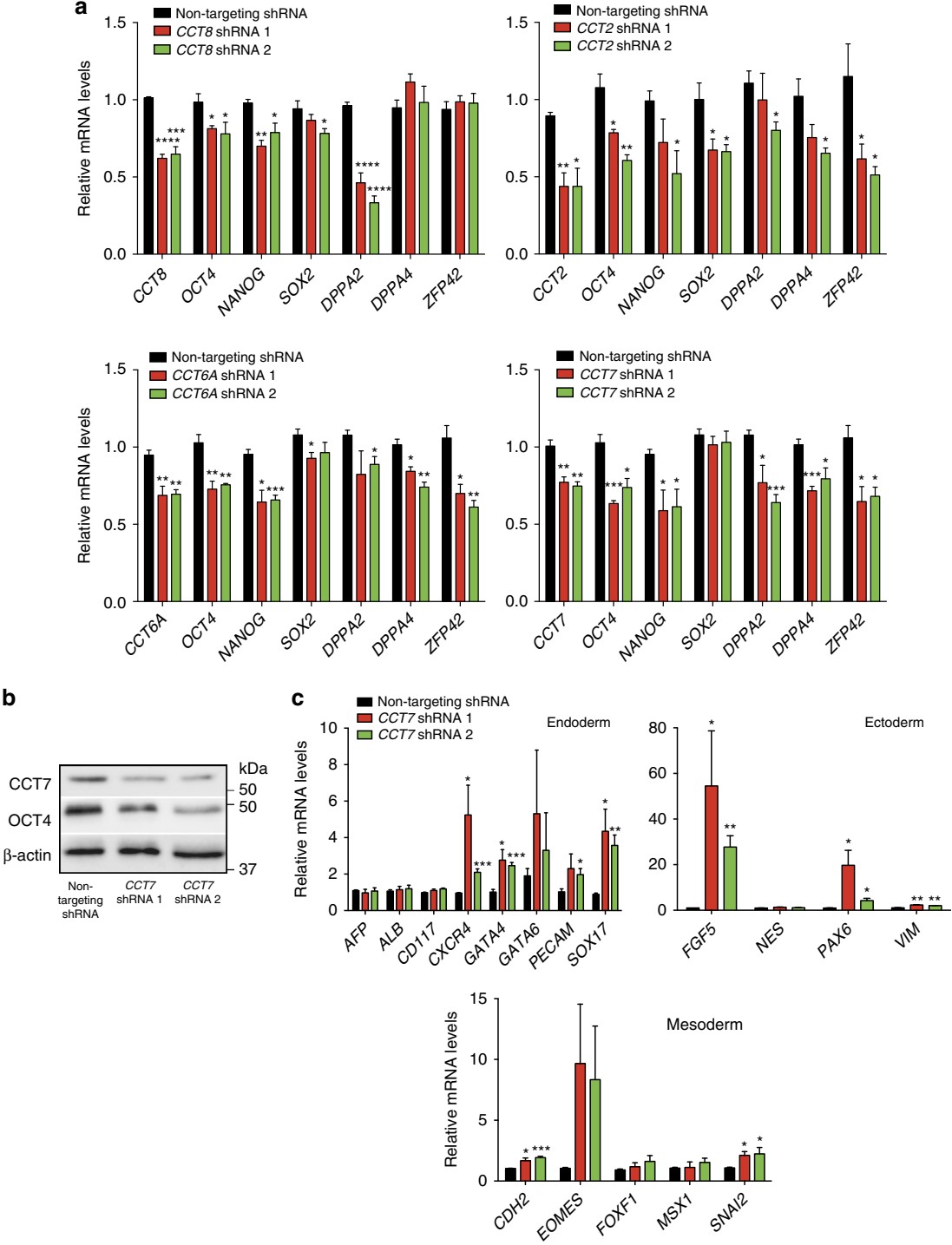

**Figure 3 | Knockdown of CCT subunits affects pluripotency of hESCs.** (**a**) Real-time PCR analysis of pluripotency markers in H9 hESCs. Graphs (relative expression to NT shRNA) represent the mean ± s.e.m. of at least three independent experiments. Knockdown of CCT8 ($n = 3$), CCT2 ($n = 3$), CCT6A ($n = 4$) and CCT7 ($n = 4$) decrease the expression of pluripotency markers. (**b**) Knockdown of CCT7 induces a decrease in OCT4 protein levels. β-actin is the loading control. The images are representative of three independent experiments. (**c**) Real-time PCR analysis of germ-layer markers in H9 hESCs (relative expression to NT shRNA). Graph represents the mean ± s.e.m. of four independent experiments. All the statistical comparisons were made by Student's $t$ test for unpaired samples. $P$ value: *$P < 0.05$, **$P < 0.01$, ***$P < 0.001$, ****$P < 0.0001$.

and mammalian cells, the inhibition of HSP90 induces heat-shock response (HSR) and reduces polyQ-expanded protein aggregation[22]. We found that the treatment with an inhibitor of HSP90 did not further decrease the signal observed in HD-iPSCs by filter trap, reinforcing that these cells do not accumulate detectable aggregates of mutant HTT (Supplementary Fig. 8a).

However, a collapse in proteostasis induced by proteasome inhibition triggered the accumulation of polyQ aggregates in HD-iPSCs (ref. 23; Supplementary Fig. 8b–c).

Loss of CCT subunits enhances aggregation of mutant HTT and worsens HD-related changes in yeast, *C. elegans* and mammalian neuronal models[24–26]. Thus, increased TRiC/CCT

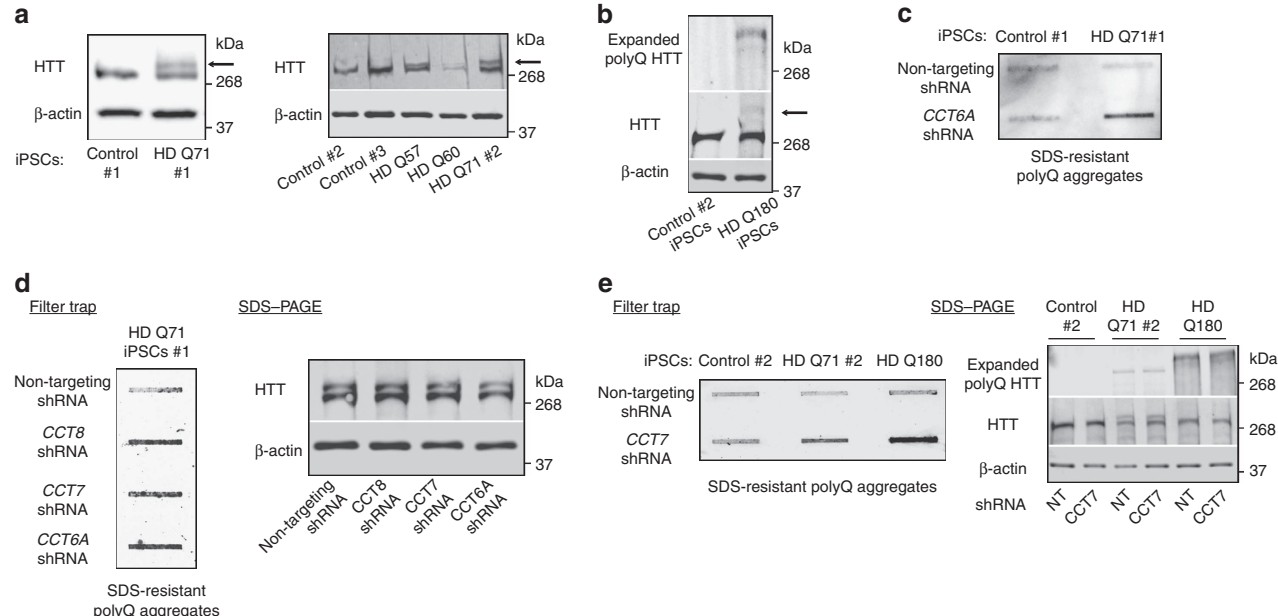

**Figure 4 | Knockdown of CCT subunits impairs proteostasis of pluripotent stem cells.** (**a**) Western blot analysis of control and HD-iPSC lysates with antibodies to HTT and β-actin. Arrow indicates mutant polyQ-expanded HTT. The images are representative of three independent experiments. (**b**) In HD polyQ180-expressing iPSCs, the levels of mutant HTT are dramatically decreased compared with normal HTT copy. The expression of polyQ180 HTT was confirmed by using an antibody that detects polyQ-expanded mutant HTT. The images are representative of three independent experiments. (**c**) Filter trap analysis shows that HD-iPSCs do not have increased levels of polyQ aggregates compared with control iPSCs (detected by anti-polyQ-expansion diseases marker antibody). However, knockdown of a single CCT subunit triggers the accumulation of polyQ aggregates. The images are representative of three independent experiments. (**d**) Knockdown of different CCT subunits in HD polyQ71 iPSC line #1 results in a similar increase of polyQ aggregates. Right panel: SDS–PAGE analysis with antibodies to HTT and β-actin loading control. The images are representative of three independent experiments. (**e**) Knockdown of a single CCT subunit triggers the accumulation of polyQ aggregates in both polyQ71 iPSC line #2 and polyQ180 iPSCs. Right panel: polyQ-expansion diseases marker antibody was used to confirm that the total protein levels of mutant HTT do not change on knockdown of CCT subunits. The images are representative of two independent experiments.

assembly induced by high levels of CCT subunits could contribute to maintain the proteostasis of mutant HTT in iPSCs. We found that knockdown of CCT8 and other CCT subunits trigger the accumulation of polyQ aggregates without affecting the mutant HTT total protein levels in HD-iPSCs (Fig. 4d,e and Supplementary Fig. 9). These results were observed in all the HD-iPSC lines tested whereas knockdown of CCT subunits did not induce accumulation of polyQ aggregates in control iPSCs (Fig. 4c–e and Supplementary Fig. 10). Remarkably, loss of different CCT subunits (that is, CCT2, CCT6A, CCT7 and CCT8) had similar effects (Fig. 4d and Supplementary Fig. 9), indicating that the TRiC/CCT complex modulates polyQ aggregation in HD-iPSCs.

HD-iPSCs represent a valuable resource to gain mechanistic insights into HD (ref. 20). Neuronal dysfunction and death occurs in many brain regions in HD, but striatal neurons expressing cyclic AMP-regulated phosphoprotein (DARPP32) undergo the greatest neurodegeneration[27]. HD-iPSCs can be terminally differentiated into neurons (including DARPP32-positive cells) that exhibit HD-associated phenotypes such as cumulative risk of death over time and increased vulnerability to excitotoxic stressors[20]. Furthermore, proteotoxicity induced via autophagy inhibition or oxidative stress results in higher neurodegeneration of HD cells compared with controls[20]. Despite these phenotypes, mutant HTT-expressing neurons present important limitations for disease modelling such as lack of robust neurodegeneration, polyQ aggregates and gene expression changes resembling HD (refs 20,23). Despite the efforts to detect polyQ aggregates under different conditions (for example, adding cellular stressors), the presence of inclusions has not been observed in neurons derived

from HD-iPSCs (refs 20,23). The lack of inclusions in these cells could reflect the long period of time before aggregates accumulate in HD (ref. 20). Consistently, HD human neurons do not accumulate detectable polyQ aggregates at 12 weeks after transplantation into HD rat models whereas these inclusions are observed after 33 weeks of transplantation[23]. To assess whether loss of CCT subunits triggers neurodegeneration and polyQ aggregation in these cells, we differentiated three HD-iPSC lines (Q57, Q71 and Q180) into striatal neurons[28]. Among those cells expressing the neuronal marker MAP2, ~50% also expressed DARPP32 (Supplementary Fig. 11a). As with proteasome inhibitor treatment, we could not detect polyQ aggregates in mutant HTT-expressing neurons on knockdown of different CCT subunits (that is, CCT2, CCT6, CCT7 and CCT8) by either immunohistochemistry (Supplementary Fig. 11b) or filter trap (Supplementary Fig. 11c). However, knockdown of CCTs resulted in increased cell death of HD neuronal cultures as assessed by activation of caspase-3 whereas it did not induce neurodegeneration of control cells (Supplementary Fig. 11d). In contrast, proteasome inhibition triggered cell death at the same extent in both control and HD neurons (Supplementary Fig. 11d). These results indicate that mutant HTT-expressing neurons are more susceptible to TRiC/CCT dysfunction than controls, providing a link between downregulation of CCTs with onset of neurodegeneration in HD during aging.

**Somatic overexpression of CCT8 extends organismal lifespan.** Our results indicate that increased TRiC/CCT complex is a key determinant of proteostasis of immortal hESCs/iPSCs. However,

the levels of CCT subunits decreased on differentiation. During human brain aging, the expression of CCTs is further repressed[11]. With the strong correlation between *cct* levels, differentiation and aging, we asked whether inducing TRiC/CCT assembly in somatic post-mitotic cells could have a positive role in longevity. To examine the impact of TRiC/CCT on organismal aging and proteotoxic resistance, we used the nematode *C. elegans*. In this organism, CCT transcripts are detected in most tissues and developmental stages[29]. The role of TRiC/CCT complex in proliferating cells during *C. elegans* development has been extensively studied[30]. Disruption of TRiC/CCT assembly by knockdown of different CCT subunits causes a variety of defects in cell division and results in embryonic lethality[30–32]. These effects are partially mediated by a collapse in microtubule function as a consequence of diminished folding of tubulin by TRiC/CCT (ref. 30). In addition, loss of different *cct* subunits during post-embryonic developmental stages results in larval arrest, body morphology alterations as well as defects in developing gonads and sterility, indicating a key role of the TRiC/CCT complex in *C. elegans* development[30]. In adult worms, the only proliferating cells are found in the germline whereas somatic tissues are formed exclusively by post-mitotic cells[33]. To examine the expression of CCTs in germ cells, we dissected the germline and intestine from adult *C. elegans* and compared the levels of CCT subunits by immunohistochemistry. We found that CCT1 subunit is enhanced in germ cells compared with the intestine (Supplementary Fig. 12a), suggesting that CCTs are highly expressed in the germline. Notably, knockdown of *cct* subunits during adulthood dramatically decreased the number of germ cells destabilizing the germline (Supplementary Fig. 12b). Accordingly, we observed a dramatic decrease in the number of laid eggs after 1 day of *cct* RNA interference (RNAi) treatment during adulthood (Supplementary Fig. 12c). Overall, these data suggest that high levels of TRiC/CCT complex are essential for proliferating cells and germline stability. However, knockdown of *cct* subunits during adulthood did not decrease lifespan in wild-type worms (Supplementary Fig. 13 and Supplementary Data 3).

To assess the expression of CCT subunits in somatic tissues of adult *C. elegans*, we generated a green fluorescent protein (GFP) transcriptional reporter construct for *cct-8* gene. Although we did not observe GFP expression in germ cells as a result of germline silencing of transgenes[34,35], these experiments confirmed wide expression of *cct-8* in somatic cells including neurons or body wall muscle cells (Supplementary Fig. 14). Somatic expression pattern of *cct-8* resembled other *cct* subunits (for example, *cct-1*, *cct-2* and *cct-7*) showing a high expression in pharynx and tail[30,35] (Supplementary Fig. 14). In aging organisms, post-mitotic somatic tissues undergo a gradual deterioration and become particularly susceptible to age-associated protein aggregation diseases. Thus, we asked whether increasing the levels of the TRiC/CCT complex in somatic tissues could delay the aging of post-reproductive organisms. To examine this hypothesis we overexpressed CCT8, a subunit that promotes TRiC/CCT assembly in mammalian cells (Fig. 2c). For this purpose, we induced ectopic expression of *cct-8* under *sur-5* promoter (Fig. 5a), which is expressed ubiquitously in somatic tissues but not in the germline[10,36]. Notably, somatic OE of *cct-8* was sufficient to extend the lifespan of *C. elegans* under normal conditions (20 °C; Fig. 5b and Supplementary Data 3). These worms exhibited up to 20% increased median lifespan compared with the control strain and other *cct*-overexpressing worms (Fig. 5b). We observed similar results in two independent OE lines of the different *cct* subunits tested (Supplementary Data 3). Since OE of a single *cct* subunit did not change the levels of other subunits, the longevity phenotype can be attributed specifically to

*cct-8* (Fig. 5a). Interestingly, the lifespan extension induced by ectopic expression of *cct-8* was more dramatic at 25 °C, a condition that results in mild heat stress (Fig. 5c). Under this temperature, *cct-8(OE) C. elegans* lived up to 40% longer than the control strain. Overexpression of *cct-2* also increased lifespan significantly at 25 °C, although to a lesser extent than *cct-8* (Fig. 5c). Since heat stress challenges the structure of proteins and triggers the accumulation of misfolded proteins, our results indicate that both *cct-8* and *cct-2* extend longevity by sustaining the integrity of the proteome during adulthood.

To examine whether these pro-longevity effects are induced through modulation of the TRiC/CCT complex, we knocked down the expression of a different subunit. Given that *cct* subunits are required during larval development, we initiated RNAi treatment during adulthood. Interestingly, knockdown of *cct-6* partially reduces the longevity phenotype of both *cct-8(OE)* and *cct-2(OE)* worms whereas it did not affect the lifespan of the control strain (Fig. 5d and Supplementary Fig. 15). By native gel electrophoresis analysis, we confirmed that OE of *cct-8* in somatic tissues increases TRiC/CCT assembly in the form of two stacked rings (Fig. 5e). Collectively, these results suggest that somatic OE of *cct-8* induces longevity via modulation of TRiC/CCT assembly, particularly under proteotoxic conditions.

**CCT8 determines proteotoxic stress resistance.** During aging, organisms lose their ability to maintain proteostasis and respond to proteotoxic stress[3,5]. With age, the expression of several *cct* subunits significantly decreased in *C. elegans* whereas the levels of *cct-1* remained similar (Fig. 6a and Supplementary Fig. 16). At day 5 of adulthood, several *cct* subunits (for example, *cct-2*, *cct-5* and *cct-8*) were already downregulated (Fig. 6a and Supplementary Fig. 16). To examine whether somatic *cct-8(OE)* ameliorates the age-associated demise in proteotoxic stress responses, we induced acute heat stress (34 °C) at different ages (that is, days 1 and 5). Severe heat stress dramatically affects survival and activates the HSR, an essential mechanism to ensure proper cytosolic protein folding and alleviate proteotoxic stress[37]. Under heat stress (34 °C), *cct-8(OE)* worms did not survive significantly longer compared with control strain at day 1 of adulthood (Fig. 6b and Supplementary Data 3). However, *cct-8(OE)* worms were markedly more resistant to proteotoxicity than control strains when subjected to heat stress at day 5 of adulthood (Fig. 6c). Although to a lesser extent, *cct-2(OE)* also conferred resistance to heat stress (Fig. 6c). As a more formal test, we asked whether animals with reduced HSR had increased survival when *cct-8* was overexpressed (Fig. 6d). Heat-shock transcription factor activates the HSR and is required for proteotoxicity resistance and adult lifespan[38]. Thus, we induced downregulation of HSR via silencing *hsf-1* expression. Notably, *hsf-1*-RNAi-treated *cct-8(OE)* worms lived markedly longer compared with control worms under the same treatment (Fig. 6d). These results indicate that *cct-8(OE) C. elegans* can significantly overcome the loss of a key transcription factor such as *hsf-1*, which is not only required for HSR but also lifespan.

Growing evidence suggest that the TRiC/CCT complex could be a therapeutic target in HD (refs 24,26). For instance, simultaneous OE of all eight CCT subunits ameliorates polyQ-expanded HTT aggregation and neuronal death[24]. Intrigued by the protection that *cct-8* could confer, we tested whether increased levels of this subunit are sufficient to protect from polyQ-expanded aggregation. For this purpose, we used a *C. elegans* model that expresses polyQ peptides throughout the nervous system. In these worms, neurotoxicity correlates with increased length of the polyQ repeat and age[8,39]. The neurotoxic effects can be monitored by worm motility, which is markedly

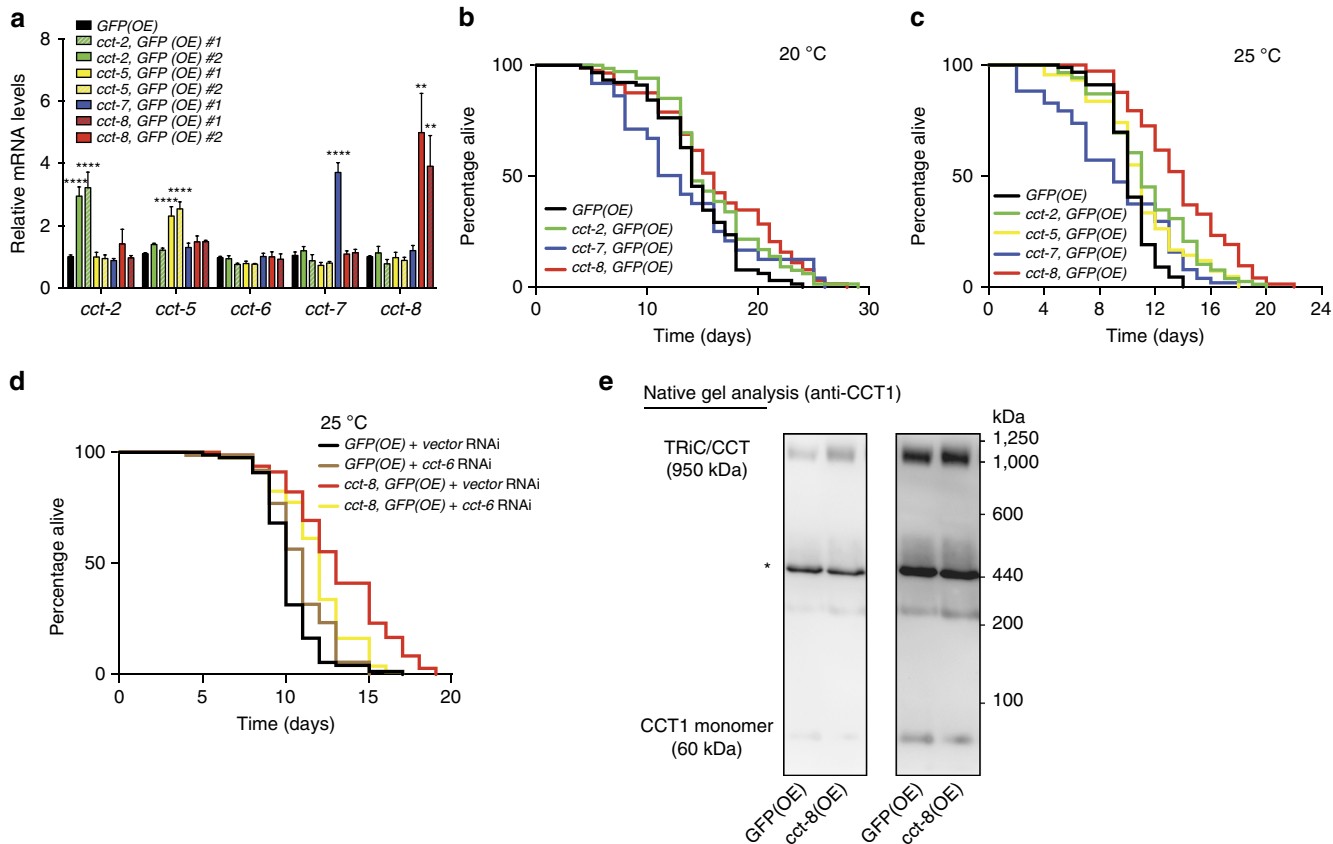

**Figure 5 | Somatic increased expression of *cct-8* induces TRiC/CCT assembly and extends longevity.** (**a**) Data represent the mean ± s.e.m. of the relative expression levels to *GFP(OE)* worms ($n = 4$ independent experiments). Statistical comparisons were made by Student's *t* test for unpaired samples. *P* value: \*\**P* < 0.01, \*\*\*\**P* < 0.001. (**b,c,d**) Each lifespan graph shows a Kaplan–Meier survival plot of a single representative experiment. In each graph, experimental and control animals were grown in parallel. $n =$ total number of uncensored animals/total number (uncensored + censored) of animals observed in each experiment. *P* values refer to experimental and control animals in a single lifespan experiment. See Supplementary Data 3 for statistical analysis, replicate data and independent OE lines results of lifespan experiments. (**b**) *cct-8(OE)* extends lifespan at 20 °C (log rank, *P* < 0.0001). *GFP(OE)*: median = 14, $n = 73/96$; *cct-2, GFP(OE)*: median = 14, $n = 65/96$; *cct-7, GFP(OE)*: median = 13, $n = 46/74$; *cct-8, GFP(OE)*: median = 16, $n = 70/96$. (**c**) Both *cct-8(OE)* and *cct-2(OE)* worms live longer compared with control *GFP(OE)* strain at 25 °C (log rank, *P* < 0.0001). *cct-8(OE)* worms live longer compared with *cct-2(OE)* and other *cct(OE)* strains (log rank, *P* < 0.0001). *GFP(OE)*: median = 10, $n = 89/96$; *cct-2, GFP(OE)*: median = 11, $n = 79/96$; *cct-5, GFP(OE)*: median = 11, $n = 42/50$; *cct-7, GFP(OE)*: median = 9, $n = 56/77$; *cct-8, GFP(OE)*: median = 14, $n = 82/96$. (**d**) Knockdown of *cct-6* reduces the long lifespan induced by *cct-8(OE)* (log rank, *P* = 0.0001). In contrast, loss of *cct-6* does not decrease the lifespan of *GFP(OE)* worms. RNAi was initiated at day 1 of adulthood. *GFP(OE)* fed *empty vector* RNAi bacteria: median = 10, $n = 74/96$; *GFP(OE)* fed *cct-6* RNAi bacteria: median = 11, $n = 73/96$; *cct-8, GFP(OE)* fed *empty vector* bacteria: median = 13, $n = 77/96$; *cct-8, GFP(OE)* fed *cct-6* RNAi bacteria: median = 12, $n = 79/96$. (**e**) Native gel electrophoresis of *GFP(OE)* and *cct-8,GFP(OE)* worm extracts followed by immunoblotting with CCT-1 antibody (two different exposure times of the same membrane are shown). Overexpression of *cct-8* induces an increase in the assembly of TRiC/CCT in the form of two stacked rings. \*indicates a complex of ∼475 kDa which is strongly detected in worm lysates. The molecular weight suggests that this signal corresponds to single ring TRiC/CCT assembled forms. The images are representative of two independent experiments.

reduced by the aggregation of polyQ peptides with a pathogenic threshold at a length of 35–40 glutamines[39]. We found that ectopic expression of *cct-8* reduced toxicity and improve motility of worms expressing polyQ67 either at 20 °C or 25 °C (Fig. 6e and Supplementary Fig. 17). Similarly, *cct-2(OE)* also ameliorated the neurotoxic effects of polyQ67 aggregation although the impact of *cct-8* is more dramatic (Fig. 6e and Supplementary Fig. 17). Moreover, filter trap analysis showed that *cct-8(OE)* reduced polyQ67 aggregates without decreasing the total protein levels of polyQ67 (Fig. 6f). In fact, we observed higher expression of total polyQ67 protein in *cct-8(OE)* worms probably because they are in a healthier state than controls. Taken together, these results indicate CCT8 as a novel candidate to sustain proteostasis during aging.

Longevity promoting pathways such as reduced insulin–insulin-like growth factor signalling (IIS) and dietary restriction

(DR) delay the onset of age-related diseases associated with proteostasis dysfunction[4]. These pathways confer increased proteome integrity and resistance to proteotoxic stress during aging[4,5], mechanisms that contribute to lifespan extension. Thus, we hypothesized that the TRiC/CCT complex could be required for the longevity phenotype induced by DR and reduced IIS. Notably, loss of function of TRiC/CCT complex significantly decreased longevity of long-lived IIS and DR genetic models whereas it did not affect the lifespan of wild-type worms (Fig. 7a). In *C. elegans*, the aging of somatic tissues is regulated by signals from the germline[40]. Concomitantly, ablation of germ cells extends lifespan and induces heightened resistance to proteotoxic stress in post-mitotic tissues[10,40]. We found that TRiC/CCT dysfunction reduces the lifespan of germline-lacking genetic worm models (Fig. 7b). We have previously reported that removal of germ cells induces increased expression of key

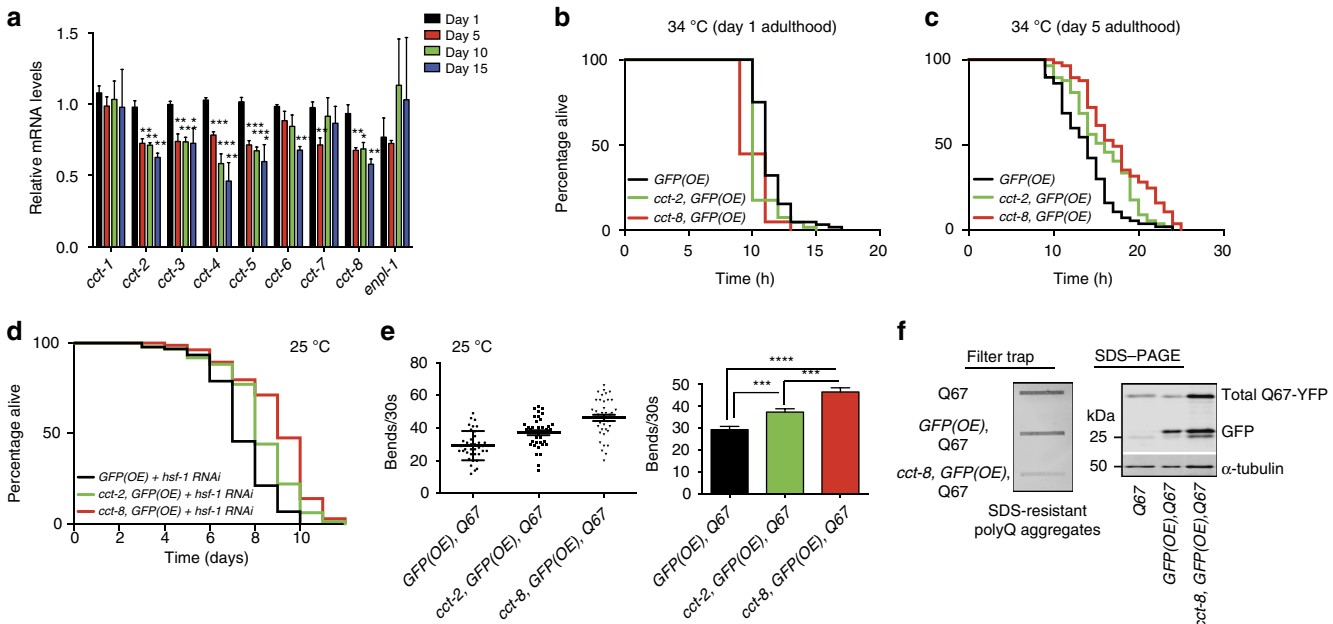

**Figure 6 | *cct-8* determines proteotoxic stress resistance during aging and protects from polyQ aggregation.** (**a**) The expression of several *cct* subunits decreases with age. In contrast, we did not find significant differences in *cct-1* and *enpl-1*, the *C. elegans* orthologue to the human chaperone *HSP90B1*. Data represent the mean ± s.e.m. of the relative expression levels to day 1 adult worms grown at 20 °C (*n* = 4 independent experiments). These experiments were performed with the sterile control strain *fer-15(b26)II;fem-1(hc17)*. (**b**) When subjected to heat stress (34 °C) at day 1 of adulthood, both *cct-8(OE)* and *cct-2(OE)* worms show similar survival rates compared with the control strain (*GFP(OE)*: median = 11, *n* = 70/72; *cct-2, GFP(OE)*: median = 10, *n* = 71/72; *cct-8, GFP(OE)*: median = 9, *n* = 70/72). (**c**) In contrast, *cct-8(OE)* (log rank, *P* < 0.0001) and *cct-2(OE)* (log rank, *P* = 0.0036) worms survive longer than control strain when subjected to heat stress at day 5 of adulthood (*GFP(OE)*: median = 14, *n* = 57/58; *cct-2, GFP(OE)*: median = 16, *n* = 57/57; *cct-8, GFP(OE)*: median = 17, *n* = 57/57). (**d**) *hsf-1*-RNAi-treated *cct(OE)* worms were long-lived compared with control strain under the same treatment (log rank, *P* < 0.0001). RNAi was initiated at day 1 of adulthood. *GFP(OE)* fed *hsf-1* RNAi bacteria: median = 7, *n* = 90/96; *cct-2, GFP(OE)* fed *hsf-1* RNAi bacteria: median = 8, *n* = 82/96; *cct-8, GFP(OE)* fed *hsf-1* RNAi bacteria: median = 9, *n* = 72/96. (**e**) Ectopic expression of *cct-8* and *cct-2* improve motility in polyQ67 worms grown at 25 °C. In the left panel, each point represents the average thrashing rate of a single 3 day-adult animal over a period of 30 s. In the right panel, bar graphs represent average ± s.e.m. of these data (*GFP(OE);Q67* (*n* = 37), *cct-2,GFP(OE);Q67* (*n* = 38), *cct-8,GFP(OE);Q67* (*n* = 38)). All the statistical comparisons were made by Student's *t* test for unpaired samples. *P* value: ***P* < 0.001, ****P* < 0.0001. (**f**) Filter trap indicates that *cct-8* OE results in decreased polyQ aggregates (detected by anti-GFP antibody). Right panel: SDS–PAGE analysis with antibodies to GFP and α-tubulin loading control. The images are representative of two independent experiments. See Supplementary Data 3 for statistical analysis and replicate data of lifespan and stress assays experiments.

proteasome subunits in somatic tissues[10]. An intriguing possibility is that the germline also modulates the levels of CCT subunits in somatic tissues. Indeed, germline-lacking *C. elegans* showed upregulation of CCT expression in somatic tissues (Fig. 7c,d). Altogether, our data suggest that the TRiC/CCT complex could be a key determinant of proteotoxic resistance and longevity.

## Discussion

Our findings establish enhanced assembly of the TRiC/CCT complex as an intrinsic characteristic of pluripotent stem cells. This increased assembly correlates with an upregulation of several CCT subunits. Notably, human pluripotent stem cells are able to maintain high expression of CCT subunits during unlimited proliferation. However, pluripotent stem cells lose their high levels of CCT subunits on differentiation and this decline is already significant in multipotent cells (for example, NPCs) before terminal differentiation. Therefore, enhanced TRiC/CCT assembly could be linked to the immortal and pluripotent characteristics of hESCs/iPSCs. Indeed, we found that slight dysfunctions of the TRiC/CCT complex induce a decrease in pluripotency markers and, concomitantly, an increase in differentiation markers. Defining the mechanisms by which increased TRiC/CCT regulates hESC/iPSC identity could open a

new door to a better understanding of pluripotency and differentiation. Although the exact mechanisms are still unknown, the TRiC/CCT complex could impinge on hESC/iPSC function in several (and non-exclusive) manners. TRiC/CCT assists the folding of a significant percentage of nascent proteins such as actin and tubulin[14,41]. Thus, one possibility is that enhanced TRiC/CCT assembly is required for the proper folding of specific regulatory or structural proteins involved either in maintenance of hESC/iPSC identity or generation of healthy differentiated cells. Interestingly, a study performed in calreticulin$^{-/-}$ mouse ESCs showed a downregulation in the expression of *Cct2*, *Cct3* and *Cct7* (ref. 42) in these cells. Calreticulin, a chaperone that binds to misfolded proteins preventing their export from the endoplasmic reticulum, is essential for cardiac development in mice[43] and required for proper myofibril formation during cardiomyocyte differentiation of mouse ESCs *in vitro*[44]. Since the TRiC/CCT complex regulates folding and actin dynamics, the downregulation of Cct subunits in calreticulin$^{-/-}$ mouse ESCs could forecast the myofibrillar disarray observed in their cardiomyocytes counterparts[42]. As TRiC/CCT also regulates the aggregation of misfolded and damaged proteins, increased TRiC/CCT levels could be linked to the biological purpose of pluripotent stem cells and their intrinsic properties such as high global protein synthesis rates and immortality. Given the ability of hESCs to replicate indefinitely

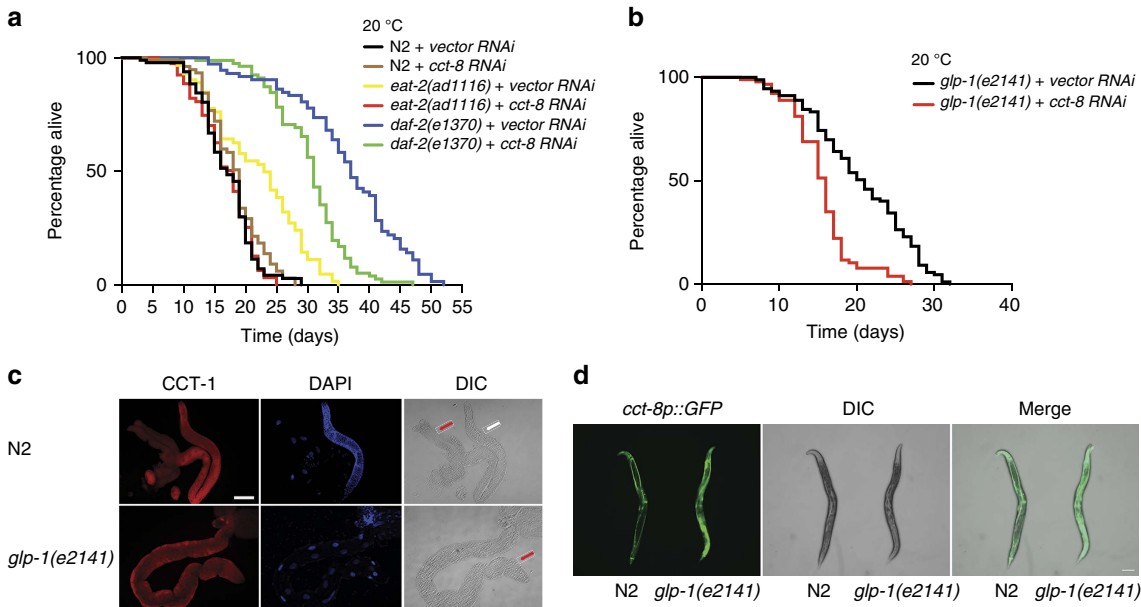

**Figure 7 | TRiC/CCT is required for the longevity phenotype of long-lived _C. elegans_ mutants.** (**a**) Knockdown of _cct-8_ decreases lifespan of IIS (_daf-2_) and DR (_eat-2_) long-lived mutant worms at 20 °C (log rank, _P_ < 0.0001). N2 (wild-type worms) + _vector_ RNAi: median = 17, _n_ = 72/9; N2 + _cct-8_ RNAi: median = 19, _n_ = 66/96; _eat-2(ad1116)_ + _vector_ RNAi: median = 23, _n_ = 66/96; _eat-2(ad1116)_ + _cct-8_ RNAi: median = 18, _n_ = 67/96; _daf-2(e1370)_ + _vector_ RNAi: median = 37, _n_ = 68/96; _daf-2(e1370)_ + _cct-8_ RNAi: median = 31, _n_ = 77/96. (**b**) Knockdown of _cct-8_ decreases lifespan of germline-lacking mutant worms (log rank, _P_ < 0.0001). _glp-1(e2141)_ + _vector_ RNAi: median = 21, _n_ = 88/96; _glp-1(e2141)_ + _cct-8_ RNAi: median = 16, _n_ = 86/96. (**c**) Gonad and intestine immunostaining with CCT-1 antibody of wild-type and germline-lacking (_glp-1(e2141)_) worms. Long-lived germline-lacking nematodes have increased levels of CCT-1 in the intestine compared with wild-type worms. Cell nuclei were stained with DAPI. White arrow indicates gonad and red arrow indicates intestine. DIC, differential interference contrast. Scale bar represents 50 μm. (**d**) Representative images of GFP expressed under control of the _cct-8_ promoter in adult wild-type and germline-lacking (_glp-1(e2141)_) worms. Scale bar represents 100 μm. In **c,d** the images are representative of three independent experiments. See Supplementary Data 3 for statistical analysis and replicate data of lifespan experiments.

and generate every other cell type in the body, the TRiC/CCT complex could be a key quality control mechanism to maintain a global intact proteome for either self-renewal or the generation of progenitor cells. In this context, the accumulation of damaged and misfolded proteins caused by proteostasis defects may affect hESC/iPSC function and immortality. With the asymmetric divisions invoked by these cells, the passage of damaged proteins to progenitor cells could also compromise development and organismal aging. In support of this hypothesis, we found that the TRiC/CCT complex determines the ability of pluripotent stem cells to maintain proteostasis of mutant huntingtin.

Another important question is how pluripotent stem cells achieve their high TRiC/CCT levels. Interestingly, specific subunits such as CCT4 are relatively more abundant compared to CCT1 in hESCs whereas this proportion is reversed in NPCs (Fig. 2b). Therefore, these subunits could become limiting assembly factors during differentiation. Despite decreasing during neural differentiation, other CCT subunits (that is, CCT8 and CCT2) were more abundant relatively to the rest of subunits in both hESCs and NPCs suggesting that they could be activators of TRiC/CCT assembly rather than limiting factors. Indeed, increasing the expression of CCT8, the most abundant subunit, triggers TRiC/CCT assembly regardless the levels of other subunits such as CCT1 in both mammalian cells and _C. elegans_. Further studies will be required to understand how CCT8 promotes TRiC/CCT assembly. A fascinating hypothesis is that CCT8 could act as a scaffold protein that triggers the interaction between the different subunits.

Our findings in human pluripotent stem cells led us to identify CCT8 as a powerful candidate to sustain proteostasis during organismal aging. The somatic induction of TRiC/CCT assembly

via CCT8 OE extends lifespan and protects from proteotoxic stress in _C. elegans_. Notably, somatic CCT8 OE can partially protect this nematode from knockdown of _hsf-1_, a transcription factor that coordinates cellular protein-misfolding responses. The precise mechanism by which the TRiC/CCT complex ameliorates the detrimental effects triggered by loss HSF1 is not yet understood. In response to proteotoxic stress, HSF1 binds heat-shock elements in the promoters of target genes and triggers their expression[45]. In mammalian cells, all the CCT subunits contain heat-shock elements and are transcriptionally activated by HSF1 (ref. 46). Thus, ectopic expression of CCT8 could ameliorate the effects induced by a decrease of CCT subunits on HSF1 knockdown. In addition, a direct regulatory interaction between TRiC/CCT activity and induction of proteotoxic stress response by HSF1 has been recently reported[47,48]. HSF1A, a chemical activator of HSF1, binds to the TRiC/CCT complex and inhibits its folding activity[47,48]. Both the inactivation of TRiC/CCT complex by HSF1A or its depletion by loss of CCT subunits induce HSF1 activity[47]. Since TRiC/CCT chaperonin interacts with HSF1 (ref. 47), TRiC/CCT could have a direct repressor role in regulating HSF1 (ref. 47). However, a decrease in TRiC/CCT activity mediated by either HSF1A or knockdown of CCT subunits can also lead to the accumulation of misfolded proteins that trigger the HSF1-induced proteotoxic response[47]. In support of this hypothesis, our results suggest that increased TRiC/CCT assembly induced by CCT8 reduces the accumulation of misfolded proteins and, therefore, ameliorates the deleterious impact of reduced HSF1-mediated signalling.

During organismal aging, loss of proteostasis in somatic tissues could contribute to the late onset of age-related diseases such as HD (refs 3,11). Hence, the decrease in the levels of several CCT

subunits during human brain aging[11] may induce TRiC/CCT dysfunction, proteotoxic stress and neurodegeneration. In support of this hypothesis, knockdown of CCT subunits hastens aggregation of mutant HTT and worsens HD-related changes in HD models[11,24–26]. Previous studies have shown that simultaneous OE of all eight CCT subunits ameliorates polyQ-expanded HTT aggregation and neuronal death[24]. Furthermore, OE of CCT1 remodels the morphology of HTT aggregates and reduces neurotoxicity[26]. Although CCT1 OE does not increase TRiC/CCT assembly, this subunit modulates the interaction between HTT and TRiC/CCT complex[26]. Since CCT8 is sufficient to increase TRiC/CCT assembly, this subunit could be a candidate to correct deficiencies in age-related diseases associated with proteostasis dysfunction. Indeed, we found that OE of CCT8 can protect from the accumulation of polyQ-expanded proteins. Recently, a link between CCT8 and mutant HTT expression was observed in an integrated genomics and proteomics study of knock-in HD mouse models expressing the human *HTT* exon1 carrying different CAG lengths (polyQ20, Q80, Q92, Q111, Q140 or Q175)[49]. Notably, highly expanded-polyQ *HTT* exon1 (Q175) induces a significant increase of CCT8 at both transcript and protein levels in the striatum of young mice (6 months) whereas other subunits were not altered[49]. In contrast, CCT8 induction was not observed in knock-in mice expressing mutant HTT with lower than 175 polyQ repeats. These findings suggest that upregulation of CCT8 could be a compensatory mechanism to protect from polyQ aggregation.

While expression of CCT subunits decrease during both differentiation and biological aging, our results indicate that mimicking proteostasis of pluripotent stem cells by modifying either the chaperome or proteasome network[8] delays the aging of somatic cells and extends organismal lifespan. Given the link between sustained proteostasis in somatic cells and healthy aging, our findings raise an intriguing question: why the levels of CCT8 decrease during differentiation if this subunit could have beneficial effects on organismal longevity and proteotoxic stress resistance? In this regard, it is important not to diminish potential detrimental effects of mimicking proteostasis of pluripotent stem cells in somatic tissues. For instance, cancer cells and pluripotent stem cells not only share their immortality features but also increased proteostasis nodes such as proteasome activity and specific chaperones[50–52]. Proteasome and chaperone levels in cancer cells are consistent with the special requirements of these cells, such as elimination of aberrant proteins. Interestingly, the expression of CCT8 is increased in gliomas and hepatocellular carcinoma whereas its knockdown induces a decrease in the proliferation and invasion capacity of these cells[53,54]. Similarly, other CCT subunits have been linked to cancer[55]. Whereas proteasome inhibitors and interventions of the chaperome network have been suggested as potential strategies for anti-cancer therapy[52,56], an abnormal activation of these mechanisms in somatic dividing cells could have the opposite effect inducing their abnormal proliferation. However, although the TRiC/CCT complex is important for the proliferation of cancer cells, this chaperonin is also required for the proper folding of p53 and, therefore, promotes tumour suppressor responses[57]. Thus, the TRiC/CCT complex could be an important factor to avoid misfolding of tumour suppressors and the increase incidence of cancer during the aging process. Besides its link with cancer, other factors could explain the decline of CCT subunits during differentiation. In support of the disposable soma theory of aging[58], a fascinating hypothesis is that downregulation of the TRiC/CCT complex during differentiation is part of an organismal genetic programme that ensures a healthy progeny whereas somatic tissues undergo a progressive demise in their homeostasis and function. Due to the limitation of nutrients in nature, organisms divide the available metabolic resources between reproduction and maintenance of the non-reproductive soma. Evolutionary pressure has been theorized to force a re-allocation of the resources to prevent or eliminate damage to the germline and progeny, while little resources are placed on the maintenance of somatic cells[58]. In support of this hypothesis, signals from the germline modulate the aging of somatic tissues and removal of the germline extends lifespan[40]. Interestingly, our results indicate that germline-lacking worms exhibit increased levels of TRiC/CCT in post-mitotic cells.

Collectively, we underscore the importance of defining proteostasis of pluripotent stem cells to uncover novel mechanisms of organismal healthspan extension. Besides CCT subunits, other chaperome components also decrease during differentiation and, therefore, could be interesting targets to be studied in the context of aging. In addition, it will be fascinating to explore whether other proteostasis components are also divergent in pluripotent stem cells and if mimicking these mechanisms in somatic cells extends organismal healthspan.

## Methods

**hESCs/iPSCs culture.** The human H9 (WA09) and H1 (WA01) hESC lines were obtained from WiCell Research Institute. The human control iPSC line #1 (hFIB2-iPS4 (ref. 59)) and HD Q71-iPSC line #1 (ref. 60) were a gift from G.Q. Daley. These iPSC lines were derived and fully characterized for pluripotency in refs 59,60. ND42242 (control iPSC line #2, Q21), ND36997 (control iPSC line #3, Q33), ND41656 (HD Q57-iPSC), ND36998 (HD Q60-iPSC), ND42229 (HD Q71-iPSC line #2) and ND36999 (HD Q180-iPSC) were obtained from NINDS Human Cell and Data Repository (NHCDR) through Coriell Institute. Parental fibroblasts GM02183 (control Q33), GM03621 (HD Q60), GM04281 (HD Q71) and GM09197 (HD Q180) were obtained from Coriell Institute whereas ND30014 (control Q21) and ND33392 (HD Q57) fibroblasts were obtained from NHCDR through RUCDR Infinite Biologics at Rutgers University. hESCs and iPSCs were maintained on Geltrex (ThermoFisher Scientific) using mTeSR1 (Stem Cell Technologies). hESCs and iPSCs colonies were passaged using a solution of dispase (2 mg ml⁻¹), and scraping the colonies with a glass pipette. The hESC and iPSC lines used in our experiments had a normal diploid karyotype as indicated by single nucleotide polymorphism (SNP) genotyping (Supplementary Fig. 18). Genetic identity of hESCs was assessed by short tandem repeat (STR) analysis. The H9 and H1 hESC lines used in our study matches exactly the known STR profile of these cells across the 8 STR loci analysed (Supplementary Table 2). No STR polymorphisms other than those corresponding to H9 and H1 were found in the respective cell lines, indicating correct hESC identity and no contamination with any other human cell line. By STR analysis, we also confirmed correct genetic identity of the iPSCs used in our study with the corresponding parental fibroblast lines when fibroblasts were available (that is, ND42242, ND36997, ND41656, ND36998, ND42229, ND36999 and HD Q71-iPSC line #1; Supplementary Table 3). All the cell lines used in this study were tested for mycoplasma contamination at least once every 3 weeks. No mycoplasma contamination was detected. Research involving hESC lines was performed with approval of the German Federal competent authority (Robert Koch Institute).

**Pan-neuronal differentiation.** Neural differentiation of H9 hESCs, H1 hESCs and iPSCs was performed using the monolayer culture protocol following the STEMdiff Neural Induction Medium (Stem Cell Technologies) method based on ref. 61. Undifferentiated pluripotent stem cells were rinsed once with PBS and then we added 1 ml of Gentle Dissociation Reagent (Stem Cell Technologies) for 10 min. After the incubation period, we gently dislodged pluripotent cells and add 2 ml of Dulbecco's Modified Eagle Medium (DMEM)-F12 + 10 μM ROCK inhibitor (Abcam). Then, we centrifuged cells at 300*g* for 10 min. Cells were resuspended on STEMdiff Neural Induction Medium + 10 μM ROCK inhibitor and plated on polyornithine (15 μg ml⁻¹)/laminin (10 μg ml⁻¹)-coated plates (200,000 cells cm⁻²). For neuronal differentiation, NPCs were dissociated with Accutase (Stem Cell Technologies) and plated into neuronal differentiation medium (DMEM/F12, N2, B27 (ThermoFisher Scientific), 1 μg ml⁻¹ laminin (ThermoFisher Scientific), 20 ng ml⁻¹ brain-derived neurotrophic factor (BDNF) (Peprotech), 20 ng ml⁻¹ glial cell-derived neurotrophic factor (GDNF) (Peprotech), 1 mM dibutyryl-cyclic AMP (Sigma) and 200 nM ascorbic acid (Sigma)) onto polyornithine/laminin-coated plates as described in ref. 8. Cells were differentiated for 1–2 months, with weekly feeding of neuronal differentiation medium.

**Striatal neuron differentiation.** iPSCs were differentiated into striatal neuron by induction with sonic hedgehog in a defined medium as reported in ref. 28. Briefly, iPSCs were detached by incubating with dispase (1 mg ml⁻¹) for 20 min. The detached colonies were cultured in suspension as free-floating embryoid bodies

(EBs) in the differentiation medium consisting of DMEM/F12, 20% knockout serum replacement, $1 \times$ MEM non-essential amino acids, 2 mM L-glutamine, and 100 μM β-Mercaptoethanol. On day 4, the medium was replaced with a neural induction medium consisting of DMEM/F12, N2 supplement, $1 \times$ MEM non-essential amino acids, 2 mM glutamine and 2 μg ml$^{-1}$ heparin. On day 7, the EBs were attached to laminin-coated substrate in a 35-mm culture petri dish and cultured in the neural induction medium. In the next week, the EBs flattened and columnar neuroepithelia organized into rosette appeared in the centre of individual colonies. On day 12, sonic hedgehog (200 ng ml$^{-1}$) was added for 14 days (until day 25). From day 26, neuroepithelial spheres were dissociated with Accutase (1 unit ml$^{-1}$, Invitrogen) at 37 °C for 5 min and placed onto polyornithine/laminin-coated coverslips in the neurobasal medium in the presence of valproic acid (VPA, 10 μM, Sigma) for 1 week, followed by a set of trophic factors, including brain derived neurotrophic factor (BDNF, 20 ng ml$^{-1}$), glial-derived neurotrophic factor (10 ng ml$^{-1}$), insulin-like growth factor 1 (10 ng ml$^{-1}$) and cyclic AMP (1 μM) (all from R&D Systems). DARPP32-expressing neurons appeared by day 35.

**Endoderm differentiation.** Endoderm differentiation of H9 hESCs was performed using STEMdiff Definitive Endoderm Kit (Stem Cell Technologies).

**Cardiomyocyte differentiation.** Cardiomyocyte differentiation was performed as described in ref. 62. Confluent H1 hESCs were dissociated into single cells with Accutase at 37 °C for 10 min followed by inactivation using two volumes of F12/DMEM. Cells were counted and 230,000 cells cm$^{-2}$ where plated in ITS medium (Corning), containing 1.25 μM CHIR 99021 (AxonMedchem) and 1.25 ng ml$^{-1}$ BMP4 (R&D), and seeded on Matrigel-coated 24-well plates. After 24 h, medium was changed to TS (transferrin/selenium) medium. After 48 h, medium was changed to TS medium supplemented with 10 μM canonical Wnt-Inhibitor IWP-2 (Santa Cruz) for 48 h. Then, medium was changed to fresh TS until beating cells were observed at days 8–10. Finally, medium was changed to Knockout DMEM (ThermoFisher Scientific) supplemented with 2% FCS, L-Glutamine and Penicillin/Streptomycin until cells were used for downstream analysis.

**SNP genotyping.** The molecular karyotype was analysed by SNP genotyping using Illumina's HumanOmniExpressExome-8-v1.2 BeadArray (Illumina, Inc., San Diego, CA, USA) at the Institute for Human Genetics (Department of Genomics, Life & Brain Center, University of Bonn, Germany). Processing was performed on genomic DNA following the manufacturer's procedures. Copy number regions were detected using the cnvPartition version 3.1.6.

**STR typing.** STR analysis of H9 and H1 hESC lines was conducted using the Promega PowerPlex 21 system (Promega Corporation) by Eurofins Genomics (Germany). We analysed loci D5S818, D13S317, D7S820, D16S539, vWA, TH01, TP0X and CSF1P0 to compare with the known STR profile of these hESC lines[63]. Genotype analysis of iPSCs lines and their parental fibroblasts was performed using the following microsatellite markers: D17S1303, D16S539, D10S526, vWA, THO1, D5S818, CSF1PO and TPOX. Fluorescently labelled PCR products were electrophoresed and detected on an automated 3730 DNA Analyzer and data were analysed using Genemapper software version 3.0 to compare allele sizes between iPSCs and their parental fibroblasts (Applied Biosystems).

**Lentiviral infection of human stem cells.** Lentivirus (LV)-non targeting shRNA control, LV-shCCT2#1 (clone number TRCN0000029499), LV-shCCT2#2 (TRCN0000029500), LV-shCCT6A#1 (TRCN0000062514), LV-shCCT6A#2 (TRCN0000062515), LV-shCCT7#1 (TRCN0000147373), LV-shCCT7#2 (TRCN0000149108), LV-shCCT8#1 (TRCN0000442235) and LV-shCCT8#2 (TRCN0000438803) in pLKO.1-puro vector were obtained from Mission shRNA (Sigma). Transient infection experiments were performed as follows. hESC/iPSC colonies growing on Geltrex were incubated with mTesR1 medium containing 10 μM ROCK inhibitor (Abcam) for 2 h and individualized using Accutase. Hundred thousand cells were plated on Geltrex plates and incubated with mTesR1 medium containing 10 μM ROCK inhibitor for 1 day. Cells were infected with 5 μl of concentrated lentivirus. Plates were centrifuged at 800$g$ for 1 h at 30 °C. Cells were fed with fresh media the day after to remove virus. After 1 day, cells were selected for lentiviral integration using 1 μg ml$^{-1}$ puromycin (ThermoFisher Scientific). Cells were collected for experimental assays after 4–6 days of infection. Alternatively, we generated stable transfected hESCs. In this case, hESC colonies growing on Geltrex were incubated with mTesR1 medium containing 10 μM ROCK inhibitor for 1 h and individualized using Accutase. Fifty thousand cells were infected with 20 μl of concentrated lentivirus in the presence of 10 μM ROCK inhibitor for 1 h. Cell suspension was centrifuged to remove virus, passed though a mesh of 40 μM to obtain individual cells, and plated back on a feeder layer of fresh mitotically inactive mouse embryonic fibroblasts in hESC media (DMEM/F12, 20% knockout serum replacement (ThermoFisher Scientific), 1 mM L-glutamine, 0.1 mM non-essential amino acids, β-mercaptoethanol and 10 ng ml$^{-1}$ bFGF (Joint Protein Central)) supplemented with 10 μM ROCK inhibitor. After a few

days in culture, small hESC colonies arose. Then, we performed 1 μg ml$^{-1}$ puromycin selection during 2 days and colonies were manually passaged onto fresh mouse embryonic fibroblasts to establish new hESC lines.

**Transfection of HEK293T cells.** HEK293T cells (ATCC) were plated on 0.1% gelatin-coated plates and grown in DMEM supplemented with 10% FCS and 1% non-essential amino acids (ThermoFisher Scientific) at 37 °C, 5% CO$_2$ conditions. Cells were transfected once they reached 80–90% confluency. 1 μg CCT8 OE plasmid (Sino Biological HG11492-UT) or pCDNA3.1 empty vector plasmid (Life Technologies V870–20) were used for transfection, using Fugene HD (Promega) following manufacturer's instructions. 24 h after transfection, the cells were treated with 500 μg ml$^{-1}$ Hygromycin B gold (Invivogen) during 16 h to select for transfected cells. After 24 h incubation in normal medium, the cells were harvested for TRiC/CCT assembly experiments.

**Immunohistochemistry.** Cells were fixed with paraformaldehyde (4% in PBS) for 20 min, followed by permeabilization (0.2% Triton X-100 in PBS for 10 min) and blocking (3% BSA in 0.2% Triton X-100 in PBS for 10 min). Cells were incubated in primary antibody overnight at 4 °C (Mouse anti-Polyglutamine-Expansion Diseases Marker Antibody (Merck Millipore, MAB1574, 1:50), Rabbit anti-Cleaved Caspase-3 (Cell Signaling, 9661, 1:400), Rabbit anti-DARPP32 (Abcam, ab40801, 1:100), Chicken anti-MAP2 (Abcam, ab5392, 1:1,000). After washing with PBS, cells were incubated in secondary antibody (Alexa Fluor 488 goat anti-mouse (ThermoFisher Scientific, A-11029, 1:1,000), Alexa Fluor 647 donkey anti-chicken (Jackson ImmunoResearch, 703-605-155, 1:1,000), Alexa Fluor 568 goat anti-Rabbit (ThermoFisher Scientific A-21067, 1:1,000)) and co-stained with 2 μg ml$^{-1}$ Hoechst 33342 (Life technologies). Finally, coverslips were covered with Mowiol (Sigma).

**mRNA sequencing.** For hESCs, NPCs and neurons (1, 2 and 4 weeks of differentiation) total RNA was extracted using RNAbee (Tel-Test Inc.). Total RNA was rRNA-depleted and libraries were prepared using TruSeq Stranded Total RNA Library Prep Kit (Illumina). RNA-seq libraries were then sequenced 100 bp single end with HiSeq 2500 System. 100 bp short reads were trimmed and quality clipped with Flexbar[64]. All remaining reads (>18 bp in length) were mapped against the human 45S rRNA precursor sequence (NR_046235) to remove rRNA contaminant reads. We used the human genome sequence and annotation (EnsEMBL 79) together with the splice-aware STAR read aligner[65] (release 2.5.1b) to map and assemble our reference transcriptome. Subsequent transcriptome analyses on differential gene and transcript abundance were carried out with the cufflinks package[66]. Supplementary Data 4 provides the statistical analysis of the transcriptome data.

**Sample preparation for label-free quantitative proteomics and analysis.** Cells were scratched in urea buffer (containing 8 M urea, 50 mM ammonium bicarbonate and $1 \times$ complete protease inhibitor mix with EDTA (Roche)), homogenized with a syringe and cleared using centrifugation (16,000$g$, 20 min). Supernatants were reduced (1 mM DTT, 30 min), alkylated (5 mM IAA, 45 min) and digested with trypsin at a 1:100 w/w ratio after diluting urea concentration to 2 M. The day after, samples were cleared (16,000$g$, 20 min) and supernatant was acidified. Peptides were cleaned up using stage tip extraction[67]. In short, peptides were eluted from C18 tips with 30 μl of 0.1% formic acid in 80% acetonitrile, concentrated in a speed vac to complete dryness and resuspended in 10 μl buffer A (0.1% formic acid). The liquid chromatography tandem mass spectrometry (LC-MS/MS) equipment consisted out of an EASY nLC 1000 coupled to the quadrupole based QExactive instrument (Thermo Scientific) via a nano-spray electroionization source. Peptides were separated on an in-house packed 50 cm column (1.9 μm C18 beads, Dr Maisch) using a binary buffer system: A) 0.1% formic acid and B) 0.1% formic acid in acetonitrile. The content of buffer B was raised from 7 to 23% within 120 min and followed by an increase to 45% within 10 min. Then, within 5 min buffer B fraction was raised to 80% and held for further 5 min after which it was decreased to 5% within 2 min and held there for further 3 min before the next sample was loaded on the column. Eluting peptides were ionized by an applied voltage of 2.2 kV. The capillary temperature was 275 °C and the S-lens RF level was set to 60. MS1 spectra were acquired using a resolution of 70,000 (at 200 $m/z$), an Automatic Gain Control target of 3e6 and a maximum injection time of 20 ms in a scan range of 300–1,750 Th. In a data dependent mode, the 10 most intense peaks were selected for isolation and fragmentation in the higher-energy collisional dissociation (HCD) cell using a normalized collision energy of 25 at an isolation window of 2.1 Th. Dynamic exclusion was enabled and set to 20 s. The MS/MS scan properties were: 17,500 resolution at 200 $m/z$, an Automatic Gain Control target of 5e5 and a maximum injection time of 60 ms. All proteomics data sets (at least 5 biological replicates per condition) were analysed with the MaxQuant software[68] (release 1.5.3.30). Spectra were searched against the 'all peptides' database from EnsEMBL release 79 (Homo_sapiens.GRCh38.pep.all.fasta). We employed the label-free quantitation mode[69] and used MaxQuant default settings for protein identification and LFQ quantification. All downstream analyses were carried out on LFQ values, which have been subjected to the variance stabilization transformation method (limma)[70]. We identified differentially abundant protein

groups by linear modelling including cell type and experimental batch as variable using limma's moderated t-statistics framework. We retain all protein groups with an adjusted $P$ value ($q$-value) of $< 0.05$. For the characterization of protein level differences in the chaperome network, we extracted the annotated human chaperome from Brehme et al.[11] and intersected this dataset with our computed protein abundance fold changes and test for differential abundance. All tables and figures were generated with custom R scripts[71]. Supplementary Data 5 provides the statistical analysis of the quantitative proteomics data comparing neurons versus hESCs and NPCs versus hESCs.

**C. elegans.** strains and generation of transgenic lines. Wild-type (N2), CF512 (*fer-15(b26)II;fem-1(hc17)IV*), CB4037 (*glp-1(e2141)III*), DA1116 (*eat-2(ad1116)II*) *C. elegans* strains were obtained from the *Caenorhabditis* Genetic Center, which is funded by NIH Office of Research Infrastructure Programs (P40OD010440). CF1041 (*daf-2(e1370)III*) was a gift from C. Kenyon. *C. elegans* were handled using standard methods[72].

For the generation of worm strains DVG48–DVG49 (N2, *ocbEx48[psur5::cct-8, pmyo3::GFP]* and N2, *ocbEx49[psur5::cct-8, pmyo3::GFP]*), a DNA plasmid mixture containing 70 ng $\mu$l$^{-1}$ of the plasmid *psur5::cct-8* and 20 ng $\mu$l$^{-1}$ pPD93_97 (*pmyo3::GFP*) was injected into the gonads of adult N2 hermaphrodite animals, using standard methods[73]. GFP-positive F$_1$ progeny were selected. Individual F$_2$ worms were isolated to establish independent lines. Following this method we generated the worm strains DVG47 (N2, *ocbEx47[psur5::cct-2, pmyo3::GFP]*), DVG50 (N2, *ocbEx50[psur5::cct-2, pmyo3::GFP]*), DVG41 (N2, *ocbEx41[psur5::cct-5, pmyo3::GFP]*), DVG58 (N2, *ocbEx55[psur5::cct-5, pmyo3::GFP]*) and DVG44 (N2, *ocbEx44[psur5::cct-7, pmyo3::GFP]*). Control worms DVG9 (N2, *ocbEx9[myo3p::GFP]*) were generated by microinjecting N2 worms with 20 ng $\mu$l$^{-1}$ pPD93_97.

AM716 (*rmIs284[pF25B3.3::Q67::YFP]*) was a gift from R. I. Morimoto. For the generation of worm strains DVG59 ([*rmIs284[pF25B3.3::Q67::YFP;ocbEx9 [myo3p::GFP]*]), DVG55 ([*rmIs284[pF25B3.3::Q67::YFP;ocbEx49[psur5::cct-8, pmyo3::GFP]*]) and DVG57 ([*rmIs284[pF25B3.3::Q67::YFP;ocbEx50[psur5::cct-2, pmyo3::GFP]*]), AM716 was crossed to DVG9, DVG49 and DVG50, respectively.

**Lifespan studies.** Synchronized animals were raised and fed OP50 *E. coli* at 20 °C until day 1 of adulthood. Then, worms were transferred onto plates with RNAi clones. 96 animals were used per condition and scored every day or every other day[74]. From the initial worm population, the worms that are lost or burrow into the medium as well as those that exhibit 'protruding vulva' or undergo bagging were censored. $n =$ total number of uncensored animals/total number (uncensored + censored) of animals observed in each experiment. Lifespans were conducted at either 20 °C or 25 °C as stated in the figure legends. For non-integrated lines DVG9, DVG41, DVG44, DVG47-50 and DVG58, GFP-positive worms were selected for lifespan studies. PRISM 6 software was used for statistical analysis to determine median and percentiles. In all cases, $P$ values were calculated using the log-rank (Mantel–Cox) method. The $P$ values refer to experimental and control animals in a single experiment. In the main text, each graph shows a representative experiment. See Supplementary Data 3 for statistical analysis and replicate data.

**Construction of cct C. elegans expression plasmids.** To construct the *cct C. elegans* expression plasmids, pPD95.77 from the Fire Lab kit was digested with SphI and XmaI to insert 3.6KB of the *sur5* promoter. The resultant vector was then digested with KpnI and EcoRI to excise GFP and insert a multi-cloning site containing KpnI, NheI, NotI, XbaI and EcoRI. T21B10.7.1 (*cct-2*), C07G2.3a.1 (*cct-5*), F01F1.8a.1 (*cct-6*) and T10B5.5a (*cct-7*) were PCR amplified from cDNA to include 5′ NheI and 3′ NotI restriction sites then cloned into the aforementioned vector. Y55F3AR.3a (*cct-8*) was amplified with 5′ NheI and 3′ EcoRI. All constructs were sequence verified.

**Construction of cct-8 expression construct.** To construct pDV077, pPD95.77 from the Fire Lab kit was digested with XbaI and XmaI. The promoter region and part of the first intron of Y55F3AR.3a (*cct-8*) was PCR amplified from N2 gDNA to include $-600$ to 100 and then cloned into the aforementioned vector using the same enzymes.

**RNAi constructs.** RNAi-treated strains were fed *E. coli* (HT115) containing an empty control vector (L4440) or expressing double-stranded RNA. *cct-2*, *cct-6*, *cct-7* and *cct-8* RNAi constructs were obtained from the Vidal RNAi library. *cct-5* and *hsf-1* RNAi constructs were obtained from the Ahringer RNAi library. All constructs were sequence verified.

**C. elegans germline and gut immunostaining.** N2 and CB4037 strains were grown at 25 °C from hatching on OP50. Worms were dissected in dissection buffer (0.2% Tween 20, 1× Egg buffer, 20 mM Sodium Azide) on a coverslip. Fixation was performed by adding formaldehyde buffer (1× Egg buffer, 0.2% Tween 20 and 3.7% formaldehyde) and animals were frozen on dry ice between the coverslip and a poly-lysine coated slide (Thermo Scientific). The coverslips were removed and the

slides were placed for 1 min at $-20$ °C in methanol, then washed twice with PBS 0.1% Tween 20. After blocking for 30 min in PBST 10% donkey serum, anti-TCP1 alpha (Abcam, ab90357) antibody was added (1:1,000) followed by overnight incubation in a humid chamber. Anti Rat secondary antibody (Alexa Fluor 546) was added (1:400) for 2 h at room temperature. Finally, slides were mounted with Precision coverslip (Roth) using DAPI Fluoromount-G (Southern Biotech). Anti-TCP1 alpha is profiled for use in *C. elegans* by Abcam. We validated by western blot analysis that anti-TCP1 alpha recognizes this subunit in worms at the correct molecular weight. Native gel experiments confirmed that anti-TCP1 alpha recognizes CCT1 in its monomeric form as well as part of the TRiC/CCT complex.

**Nuclear staining in RNAi-treated C. elegans.** N2 wild-type worms were grown on *E. coli* (HT115) containing an empty control vector (L4440) from hatching until adulthood. The animals were then either kept on empty vector bacteria or fed RNAi bacteria to *cct-2* or *cct-8* until day 4 of adulthood. Animals were then harvested and washed three times in M9 buffer. After 1 h incubation in methanol, worms were washed again three times with M9 buffer and mounted on a slide using DAPI Fluoromount-G (Southern Biotech).

**Heat stress assays.** For day 1 adulthood heat-shock assays, eggs were transferred to plates seeded with *E. coli* (OP50) bacteria and grown to day 1 of adulthood at 20 °C. Worms were then transferred to fresh plates with *E. coli* (HT115) containing L4440 and heat shocked at 34 °C. Worms were checked every hour for viability. For day 5 adulthood heat-shock assays, adult worms were transferred to fresh plates with *E. coli* (HT115) containing L4440 every day and then heat shocked at 34 °C at day 5. PRISM 6 software was used for statistical analysis and $P$ values were calculated using the log-rank (Mantel–Cox) method.

**Motility assay.** Thrashing rates were determined as described in ref. 39. Animals were grown at 20 °C until L4 stage and then grown at 20 °C or 25 °C for the rest of the experiment. Worms were fed with *E. coli* (OP50) bacteria. Worms transferred at day 3 of adulthood to a drop of M9 buffer and after 30 s of adaptation the number of body bends was counted for 30 s. A body bend was defined as change in direction of the bend at the midbody.

**Western blot.** Cells were collected from tissue culture plates by cell scraping and lysed in protein cell lysis buffer (10 mM Tris-HCl, pH 7.4, 10 mM EDTA, 50 mM NaCl, 50 mM NaF, 1% Triton X-100, 0.1% SDS supplemented with 2 mM sodium orthovanadate, 1 mM phenylmethylsulphonyl fluoride and complete mini protease) for 1 h at 1,000 r.p.m. and 4 °C in a Thermomixer. Samples were centrifuged at 10,000$g$ for 15 min at 4 °C and the supernatant was collected. Protein concentrations were determined with a standard BCA protein assay (Thermo Scientific). Approximately 20–30 $\mu$g of total protein was separated by SDS–polyacrylamide gel electrophoresis (SDS–PAGE), transferred to nitrocellulose membranes (Whatman) and subjected to immunoblotting. Western blot analysis was performed with anti-TCP1 alpha (Abcam, ab90357, 1:1,000), anti-CCT2 (Abcam, ab92746, 1:10,000), anti-CCT6A (Abcam, ab140142, 1:1,000), anti-CCT7 (Abcam, ab170861, 1:10,000), anti-CCT8 (Proteintech, 12263-1-AP, 1:1,000), anti-OCT4 (Stem Cell Technologies, #60093, 1:5,000), anti-HTT (Cell Signaling, ab#5656, 1:1,000), anti-polyQ-expansion diseases marker antibody (Millipore, MAB1574, 1:1,000), anti-GFP (ImmunoKontact, 210-PS-1GFP, 1:5,000), anti-$\alpha$-tubulin (Sigma, T6199, 1:5,000), anti-GAPDH (Abcam, ab9484, 1:5,000) and anti-ß-actin (Abcam, ab8226, 1:5,000). In *C. elegans*, we tested all the CCT subunit antibodies indicated above but only anti-TCP1 alpha (Abcam, ab90357) was positive (as already profiled for use in this organism by Abcam). Uncropped versions of all important western blots are presented in Supplementary Fig. 19.

**RNA isolation and quantitative RT–PCR.** For human cell samples, total RNA was extracted using RNABee (Tel-Test Inc.). cDNA was generated using qScript Flex cDNA synthesis kit (Quantabio). SybrGreen real-time qPCR experiments were performed with a 1:20 dilution of cDNA using a CFC384 Real-Time System (Bio-Rad) following the manufacturer's instructions. Data were analysed with the comparative 2$\Delta\Delta C_t$ method using the geometric mean of *ACTB* and *GAPDH* as housekeeping genes. For *C. elegans* samples, total RNA was isolated from synchronized populations of ~2,000 adults using QIAzol lysis reagent (Qiagen). Data were analysed with the comparative 2$\Delta\Delta C_t$ method using the geometric mean of *cdc-42* and *pmp-3* as endogenous control[75]. See Supplementary Tables 4 and 5 for details about the primers used for this assay.

**Filter trap.** Worms were grown at 20 °C until L4 stage and then grown at 25 °C for the rest of the experiment. Day 3 adult worms were collected with M9 buffer and worm pellets were frozen with liquid N2. Frozen worm pellets were thawed on ice and worm extracts were generated by glass bead disruption on ice in non-denaturing lysis buffer (50 mM Hepes pH 7.4, 150 mM NaCl, 1 mM EDTA, 1% Triton X-100) supplemented with EDTA-free protease inhibitor cocktail (Roche). Worm and cellular debris was removed with 8,000$g$ spin for 5 min. Approximately 100 $\mu$g of protein extract was supplemented with SDS at a final concentration of 0.5%

and loaded onto a cellulose acetate membrane assembled in a slot blot apparatus (Bio-Rad). The membrane was washed with 0.1% SDS and retained Q67-GFP was assessed by immunoblotting for GFP (ImmunoKontakt, 210-PS-1GFP, 1:5,000). Extracts were also analysed by SDS–PAGE to determine protein expression levels.

iPSCs were collected in non-denaturing lysis buffer supplemented with EDTA-free protease inhibitor cocktail and lysed by passing 10 times through a 27 gauge needle attached to a 1 ml syringe. Then, we followed the filter trap protocol described above. The membrane was washed with 0.1% SDS and retained polyQ proteins were assessed by immunoblotting for anti-polyQ-expansion diseases marker antibody (Millipore, MAB1574, 1:5,000). This antibody recognizes polyQ-containing proteins like HTT and detects remarkably better polyQ-expanded HTT than wild-type HTT (refs 20,21). Extracts were also analysed by SDS–PAGE to determine HTT protein expression levels.

**Blue native gel immunoblotting of TRiC/CCT complex.** For experiments with hESCs (H9 and H1) and their NPC counterparts, cells were collected in lysis buffer (50 mM Tris-HCl (pH 7.5), 1 mM dithiothreitol and 10% glycerol supplemented with protease inhibitor cocktail (Roche)) and lysed by passing 10 times through a 27 gauge needle attached to a 1 ml syringe. Lysate was centrifuged at 16,000$g$ for 10 min at 4 °C. 50 μg of total protein was run on a 3–13% gel in deep blue cathode buffer (50 mM Tricine, 7.5 mM Imidazole and 0.02% Coomassie G250) at 4 °C for 3 h at 100 V and then exchange deep blue cathode buffer to slightly blue cathode buffer (50 mM Tricine, 7.5 mM Imidazole, and 0.002% Coomassie G250) and run at 100 V overnight. Proteins were then transferred to a polyvinylidene difluoride membrane at 400 mV for 3 h by semi-dry blotting. Western blot analysis was performed with anti-TCP1 alpha (Abcam, ab90357, 1:1,000), anti-CCT2 (Abcam, ab92746, 1:10,000) and anti-CCT8 (Proteintech, 12263-1-AP, 1:1,000). Extracts were also analysed by SDS–PAGE to determine CCT subunit expression levels and loading control. Uncropped versions of all important native and SDS–PAGE gels are presented in Supplementary Fig. 19.

**Data availability.** The mass spectrometry proteomics data have been deposited to the ProteomeXchange Consortium via the PRIDE partner repository under accession code PXD005123. Transcriptome data have been deposited in the Sequence Read Archive (SRA) under the accession code SRP091319. All the other data are also available from the corresponding author upon request.

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

## Acknowledgements

We thank D. Bartsch and L. Kurian for their suggestions and performing mesoderm differentiation. We thank T. Langer for his advice on the study of TRiC/CCT assembly. The following funding sources supported this research: Deutsche Forschungsgemeinschaft (DFG) (VI742/1-1 and CECAD), University of Cologne Advanced Postdoc Grant, Else Kröner-Fresenius-Stiftung (2015_A118) and European Research Council (ERC Starting Grant-677427 StemProteostasis). Proteomics experiments were performed at the CECAD Proteomics Facility. Some of the nematode strains used in this work were provided by the *Caenorhabditis* Genetics Center (University of Minnesota), which is supported by the NIH Office of Research Infrastructure Programs (P40 OD010440).

## Author contributions

A.N. and A.K. performed most of the experiments, data analysis and interpretation through discussions with D.V., R.G.-G. and S.K. performed filter trap experiments and helped with other experiments. H.J.L. and T.K. performed TRiC/CCT assembly experiments. C.S. helped with and performed some of the experiments. I.S. carried out the proteomics and RNA sequencing experiments. A.F. performed neuronal differentiation and analysis of HD-iPSC lines. C.D. performed bioinformatic analysis of proteomics and transcriptomics data. D.V. planned and supervised the project. The manuscript was written by D.V. All the authors discussed the results and commented on the manuscript.

## Additional information

**Competing financial interests:** The authors declare no competing financial interests.

