## [Peer Review File · Nature Communications]

Parts of this peer review file have been redacted as indicated to maintain the confidentiality of an ongoing research project.

Reviewer #1 (Remarks to the Author)

This manuscript submitted by Noormohammadi and colleagues identifies a novel regulatory pathway keeping stem cells more 'youthful' than other cells. The authors demonstrate that hESCs have a higher abundance of specific CCT subunits (3, 4, 5, 6A, and 8) than differentiated cells. More strikingly, hESCs appear to have a substantially higher levels of assembled CCT as compared to differentiated cells. Decreasing CCT subunit levels promotes hESC differentiation, suggesting that an enhanced protein quality control network is a necessary condition for maintenance of pluripotency.

The authors also demonstrate that the relationship between CCT levels and youthfulness is a generalizable phenomenon, in that it extends *C. elegans* lifespan as well, in particular under stressful conditions.

Finally, in an additional interesting finding, the authors show that there is a clear mechanistic link between enhanced CCT assembly in hESCs and *C. elegans* and enhanced proteostasis - in that hESCs and organisms become less sensitive to polyQ-mediated aggregation.

I think this manuscript to be of the highest quality and novelty. It is of high importance to the protein folding field, and it seems to me to be the first interesting finding about CCT in decades. The main thesis of the article, i. e. the link between CCT subunit levels and 'youthfulness' are extremely interesting. What I particularly like is that in addition to its main discoveries, this article also sheds light on a topic that has been somewhat confusing in the field for some time - namely the 'moonlighting' functions of CCT subunits. The data here would suggest that modulating the expression of subunits in isolation may actually modulate the levels of assembled rings, as opposed to simply the level of a given subunit.

Given the highly important, mechanistic, and novel aspects of the manuscript I would strongly recommend publishing it. I have some minor comments, but I view them as optional in light of space limitations.

1. I think that it would be very valuable to see a comparison between the levels of different subunits (as compared to other subunits) in hESCs/differentiated cells. This would illuminate the issue of whether certain subunits are limiting. At the very least this question is worth discussing.
2. It would be useful to discuss the issue of why CCTs and proteostasis in general are downregulated upon differentiation. Why aren't all cells pumping out CCT8 if its so useful to have? Are there downsides to having more assembled CCT?
3. The first line of the abstract is a little confusing. I think that the authors mean that hESCs can replicate continuously while avoiding senescence, not - if not for senescence they would replicate continuously, which is how it reads now.

Reviewer #2 (Remarks to the Author)

Somatic increase of CCT8 mimics proteostasis of human embryonic stem cells and extends organismal longevity in *C. elegans*

Alireza Noormohammadi^{1†}, Amirabbas Khodakarami^{1†}, Ricardo Gutiérrez-García¹, Hyun Ju Lee¹,

Tim König¹, Seda Koyuncu¹, Isabel Saez¹, Christoph Dieterich² and David Vilchez^{1*}

A. Summary of the key results

In this paper the authors have for the first time shown that human pluripotent stem cells have greater levels and assembly of chaperonin complex, TRiC/CCT, than differentiated cell types. These complexes are required at some level for maintaining pluripotency in human PSCs. Perturbing this complex results either in the cell death or decreased pluripotency levels. They also show some evidence of how these enhanced proteostatic capabilities of PSCs may be useful in age-related disease modeling with iPSCs, particularly for a neurodegenerative disease like Huntington's. They then complement their human PSC data using *C. elegans* model system establishing that proteostatic is also key for self-renewal potential of human ESCs as well as in aging. The advances described in this study are worthy of publication in Nature Communications after some additional data and revisions suggested below.

B. Originality and significance: if not novel, please include reference

This study is of great significance and novelty is high. The only other study that has shown that this complex has importance in ESC self-renewal potential is one that use murine embryonic stem cells and actually evaluates calreticulin-deficient mouse ESCs. Another relevant study looks at HSF1 in yeast-based therapeutic screen for Huntington's Disease and discovers TriC/CCT complex interaction with HSF1.

Faustino RS et al. Stem Cells 2010. Decoded Calreticulin-Deficient Embryonic Stem Cell Transcriptome Resolves Latent Cardiophenotype.

Neef DQ et al. PLoS Biol 2010. Modulation of heat shock transcription factor 1 as a therapeutic target for small molecule intervention in neurodegenerative disease.

C. Data & methodology: validity of approach, quality of data, quality of presentation

The overall quality and style of presenting the data is and approach taken is valid. Experiments have been carried out with precision. The weakness in this study is related to the HD iPSC modeling portion of the paper in Figure 2 where they are trying to show an application related to the importance of knockdown of CCT subunits in Huntington's disease and accumulation of poly Q aggregates. At least CAG repeats (> 34 CAG repeats) allelic series of from 3 different HD patient donors and their iPSC cell lines are needed as true biological replicates for each group to reach statistically sound conclusions. These HD iPSC lines are now easily available from multiple repositories. The authors could also bolster their conclusions by using multiple clonal HD iPSC lines from the CAG repeat donor. This is a norm for iPSC disease modeling field now. Moreover, the CAG repeat length for the HD patient donor iPSC shown in this paper is not described at all.

Most importantly, this paper has NOT evaluated a single phenotype or molecular mechanisms of proteotoxic stress in the relevant neural cell types from the HD iPSCs i.e. the striatal neurons that degenerate in human HD patients. The HD hESCs and hiPSCs studies published thus far have shown no phenotypes. Therefore, it is important to demonstrate relevance of disturbing the proteostasis and poly Q aggregates in a relevant neuronal cell type. Protocols for striatal neuron differentiation from iPSCs have been now published by multiple labs. PMID: 22748968; PMID: 18922775

This is necessary. There is also a lack of in-lab HD-iPSC pluripotency validation data described in supplementary figures. Even though iPSCs may be procured from another lab, basic iPSC QC within the lab is required to ensure stable cytogenetics. G-band karyotype (cytogenetic stability) and STR identity analysis is required on all the PSCs used in this study including the H1 and H9 hESCs, as well as the HD-iPSCs.

D. Appropriate use of statistics and treatment of uncertainties

This was one of the major weaknesses in this manuscript. Then n's are mentioned in each figure legend, however, it is not clearly described where the n's are derived from? Are these true biological and experimental replicates or only technical replicates? Is it independent experiments with multiple biological replicates? The statistics should be run on minimum of 3 exclusively performed independent experiments (each experiment containing multiple biological and technical replicates).

Second, it wasn't clear in the figure legends of figure 1 and 2 where and how the statistics on the data of the knock down of CCT subunits is derived from. Like, n=8, 9, and 9 for CCT2, CCT6A,

CCT7. Is that 8-9 different wells within one single experiment? Or are these true independent experimental replicates of 3-5 wells, 8-9 different times? Minimum of three biological replicates would be required in 3 independent PSC lines in 1 experiment. Then the assay and phenotype data should be averaged for the group. Then each assay has to be repeated a minimum of 3 times (for every hESC/hiPSC line) in the same experiment. Then the data should be averaged and compared across the groups. Multiple wells in 1 experiment do not qualify as independent experiments. For the HD iPSCs, a minimum n of 3-4 patients is required per group for comparison, which is now the standard for iPSC disease models being accepted and published in high impact journals. The authors also need to have an equal number of control subject iPSC lines.

In figures 3 and 4 where *C. elegans* data is presented and the n's are described as 77/96 and so on, please clearly describe is this an aggregate (meta-analysis) of the n's over multiple experiments and same/different stocks. If so, how many?

Given the significant variability in differentiation of iPSC-derivatives from experiments a minimum of 3 experimental replicates are required and data averaged over those experiments.

E. Conclusions: robustness, validity, reliability

For the many specific experiments conducted the conclusions are valid. However, there is a lack of clarity regarding the significance and role of each of the subunits in human vs *C. elegans* data.

In Figure 1, in human cells it is clear the CCT8, CCT7, CCT6A and CCT2 are important for pluripotency in hESCs and assembly of TRiC/CCT complex. Here CCT1 does not seem to be critical in this process.

In Figure 2, in human cells it is clear that knockdown CCT7, CCT6A and CCT2 affects pluripotency and proteostasis. It is not clearly explained why CCT8 drops out of these experiments?

In Figure 3 and 4, in *C. elegans*, it cct-8 are the most important subunit in increasing longevity and proteotoxic stress. The human ortholog of the *c. elegans* cct-8, CCT8 does not appear to be as critical in affecting proteotoxic stress. This rationale for this should be explained.

As a result of the discrepancies in the results between the different CCT subunits across the multiple species it is very difficult to follow the paper. Importantly, the reliability of the phenotypes cannot be ascertained due to the minimal number of PSCs utilized in each experiment group.

Suggested improvements: experiments, data for possible revision

1. Figure legend 2C. Knock down of CCT6A and CCT7 does not appear to induce expression of endoderm markers at all. Rather it should say ectoderm. Please correct this. Also, it is rather curious as to why the knockdown does not affect mesoderm and endoderm expression in 2C. The authors need to explain clearly the rationale behind this?

2. Clarify the statistics as stated previously.

3. Evaluate and include publicly available transcriptome data regarding levels and relevance of the chaperome. There is also GEO data from published HD iPSC studies that have shown some or no phenotypes in the differentiated neurons.

4. Incorporate allelic series of CAG repeat HD iPSC lines.

5. Clarify differences in species and different CCT subunits and their relevance to proteotoxic stress.

6. It would be ideal if the authors can include levels CCT subunit proteins of WT and HD BAC mice primary striatal neurons.

7. Explain why CCT6A and CCT7 knockdown is mostly ineffective in mesodermal and endodermal gene expression. Perhaps include a larger set of germ layer specific genes.

8. Show other relevant controls of inducing proteotoxic stress in human ESC/iPSCs and differentiated neurons with inhibition of proteasomes (bortezomib) or inhibition of Hsp90 (geldanamycin)

F. References: appropriate credit to previous work?

References have been appropriately cited except for some of the citations mentioned in this review.

G. Clarity and context: lucidity of abstract/summary, appropriateness of abstract, introduction and conclusions

Overall, the paper is apprehensible and well-written including the abstract. Introduction is sound, although a more clear description of the difference and relevance of the CCT subunits in engendering proteotoxic stress across human cells and *C. elegans* should be clarified better. The

conclusions and their interpretations about many of in vitro results with HD iPSCs are much overstated without additional HD iPSC donor lines.

Reviewer #3 (Remarks to the Author)

The observations that TRiC/CCT complexes are important for hESC pluripotency and can mildly increase lifespan when overexpressed in *C. elegans* are interesting, and it is always nice to see the combination of mammalian and invertebrate mechanistic experiments in one study. However, there is some convoluted logic that enters the paper with the introduction of *C. elegans* to the story (see 2-5 below):

1. Perhaps I missed it, but do TRiC/CCT components decline with passage, increasing senescence, and loss of pluripotency (not just differentiation, as shown)? This would be an important indication of the ability of TRiC/CCT components to maintain "stemness".
2. The observation that germline-less worms, particularly *glp-1* mutants, have lower levels of CCT components is odd and undermines the authors' premise, as *glp-1* worms are long-lived, yet these data suggest that CCT components are dispensable for this long lifespan. Please explain.
3. Furthermore, the lower level of cct subunits in germline-less animals does NOT suggest that the cct subunits are expressed in germ cells--there is an immense amount of signaling between germline and somatic cells, so cct subunits could be expressed in somatic cells and regulated by germline proliferation status. No such conclusion can be drawn about location of CCT subunit expression from whole-animals experiments shown in Fig. S6. This line of reasoning must be deleted, and furthermore, the location of cct expression should be identified.
4. The connection between hESC and *C. elegans* is further strained by the fact that the authors suggest that TRiC/CCT is required for "stemness" but the tissues assayed in *C. elegans*--that is, those that determine longevity--are all differentiated and post-mitotic.
5. What, if any, is the role of TRiC/CCT in the mitotic germline of *C. elegans*? This would seem to be the more natural correlate of hESCs.

Reviewer #1 (Expert in protein folding)

This manuscript submitted by Noormohammadi and colleagues identifies a novel regulatory pathway keeping stem cells more 'youthful' than other cells. The authors demonstrate that hESCs have a higher abundance of specific CCT subunits (3, 4, 5, 6A, and 8) than differentiated cells. More strikingly, hESCs appear to have a substantially higher levels of assembled CCT as compared to differentiated cells. Decreasing CCT subunit levels promotes hESC differentiation, suggesting that an enhanced protein quality control network is a necessary condition for maintenance of pluripotency.

The authors also demonstrate that the relationship between CCT levels and youthfulness is a generalizable phenomenon, in that it extends *C. elegans* lifespan as well, in particular under stressful conditions.

Finally, in an additional interesting finding, the authors show that there is a clear mechanistic link between enhanced CCT assembly in hESCs and *C. elegans* and enhanced proteostasis - in that hESCs and organisms become less sensitive to polyQ-mediated aggregation.

I think this manuscript to be of the highest quality and novelty. It is of high importance to the protein folding field, and it seems to me to be the first interesting finding about CCT in decades. The main thesis of the article, i. e. the link between CCT subunit levels and 'youthfulness' are extremely interesting. What I particularly like is that in addition to its main discoveries, this article also sheds light on a topic that has been somewhat confusing in the field for some time - namely the 'moonlighting' functions of CCT subunits. The data here would suggest that modulating the expression of subunits in isolation may actually modulate the levels of assembled rings, as opposed to simply the level of a given subunit.

Given the highly important, mechanistic, and novel aspects of the manuscript I would strongly recommend publishing it. I have some minor comments, but I view them as optional in light of space limitations.

1. I think that it would be very valuable to see a comparison between the levels of different subunits (as compared to other subunits) in hESCs/differentiated cells. This would illuminate the issue of whether certain subunits are limiting. At the very least this question is worth discussing.

This is an excellent suggestion that led us to a better understanding of the mechanism by which CCT8 modulates TRiC/CCT assembly. Since the levels of CCT1 are similar between hESCs and NPCs, we have performed the comparison between the different subunits relative to CCT1. We have now included this comparison in the manuscript (Fig. 2b). Our results suggest that specific CCT subunits (e.g., CCT4) could become limiting assembly factors during differentiation. This comparison also suggests that specific CCT subunits could have a role as assembly activators. Interestingly, CCT8

is the most abundant subunit in both hESCs and NPCs indicating that CCT8 is not a limiting factor even during differentiation. Thus, an increase in CCT8 total levels could trigger TRiC/CCT assembly. Indeed, we found that ectopic expression of CCT8 is sufficient to increase the assembly of TRiC/CCT in both mammalian cells (**Fig. 2c**) and *C. elegans* somatic tissues (**Fig. 5e**). These results could explain how CCT8 overexpression in post-mitotic tissues is sufficient to extend longevity and delay proteotoxic dysfunction. After including this comparison suggested by Reviewer #1 and the new assembly experiments, the link between our hESC and *C. elegans* experiments is more clear and the manuscript is easier to follow. The text now says: “Since all the subunits are required for TRiC/CCT function²⁰, downregulated CCT subunits could become structural limiting factors during differentiation and modulate the decrease of TRiC/CCT assembly. An intriguing possibility is that specific subunits can also function as assembly activators. A comparison between the levels (relative to CCT1) of the different subunits in hESCs and NPCs showed that CCT8 is the most abundant subunit in both cell types despite decreasing during differentiation (**Fig. 2b**). These findings indicate that CCT8 is not stoichiometric limiting for TRiC/CCT assembly. Thus, we asked whether an increase in the total protein levels of CCT8 could trigger TRiC/CCT assembly. Strikingly, ectopic expression of CCT8 induced an increase in TRiC/CCT assembly whereas the total protein levels of CCT1 remained similar (**Fig. 2c**). In contrast, we found a decrease in the levels of CCT1 in its monomeric form upon CCT8 overexpression (**Fig. 2c**). Collectively, our results suggest that hESCs have an intrinsic proteostasis network characterized by high levels of TRiC/CCT complex. However, the levels of several CCT subunits decrease during differentiation resulting in diminished assembly of TRiC/CCT chaperonin. In addition, we identified CCT8 as a potential activator of TRiC/CCT assembly by using hESCs as a model to study proteostasis”.

Furthermore, we discussed in more detail potential mechanisms by which CCT8 could activate TRiC/CCT assembly and the potential role of other subunits as stoichiometric limiting factors during differentiation. The Discussion section now says: “Another important question is how hESCs achieve their high TRiC/CCT levels. Interestingly, specific subunits such as CCT4 are relatively more abundant compared to CCT1 in hESCs whereas this proportion is reversed in NPCs (**Fig. 2b**). Therefore, these subunits could become limiting assembly factors during differentiation. Despite decreasing during neural differentiation, other CCT subunits (i.e., CCT8 and CCT2) were more abundant relatively to the rest of subunits in both hESCs and NPCs suggesting that they could be activators of TRiC/CCT assembly rather than limiting factors. Indeed, increasing the expression of CCT8, the most abundant subunit, triggers TRiC/CCT assembly regardless the levels of other subunits such as CCT1 in both mammalian cells and *C. elegans*. Further studies will be required to understand how CCT8 promotes TRiC/CCT assembly. A fascinating hypothesis is that CCT8 could act as a scaffold protein that triggers the interaction between the different subunits”.

2. It would be useful to discuss the issue of why CCTs and proteostasis in general are downregulated upon differentiation. Why aren't all cells pumping out CCT8 if its so useful to have? Are there downsides to having more assembled CCT?

We agree with reviewer #1 that this is an intriguing question. We have now discussed the potential downsides of having more assembled TRiC/CCT complex. In short, we have commented on the relevance of the TRiC/CCT complex for cancer cell proliferation and also its potential impact on the disposable soma theory of aging. The text now says: "While expression of CCT subunits decrease during both differentiation and biological aging, our results indicate that mimicking proteostasis of hESCs by modifying either the chaperome or proteasome network¹⁰ delays the aging of somatic cells and extends organismal lifespan. Given the link between sustained proteostasis in somatic cells and healthy aging, our findings raise an intriguing question: why the levels of CCT8 decrease during differentiation if this subunit could have beneficial effects on organismal longevity and proteotoxic stress resistance? In this regard, it is important not to diminish potential detrimental effects of mimicking proteostasis of ESCs in somatic tissues. For instance, cancer cells and ESC not only share their immortality features but also increased proteostasis nodes such as proteasome activity and specific chaperones⁵⁴⁻⁵⁶. Proteasome and chaperone levels in cancer cells are consistent with the special requirements of these cells, such as elimination of aberrant proteins. Interestingly, the expression of CCT8 is increased in gliomas and hepatocellular carcinoma whereas its knockdown induces a decrease in the proliferation and invasion capacity of these cells^{57,58}. Similarly, other CCT subunits have been linked to cancer⁵⁹. Whereas proteasome inhibitors and interventions of the chaperome network have been suggested as potential strategies for anticancer therapy^{56,60}, an abnormal activation of these mechanisms in somatic dividing cells could have the opposite effect inducing their abnormal proliferation. However, although the TRiC/CCT complex is important for the proliferation of cancer cells, this chaperonin is also required for the proper folding of p53 and, therefore, promotes tumor suppressor responses⁶¹. Thus, the TRiC/CCT complex could be an important factor to avoid misfolding of tumor suppressors and the increase incidence of cancer during the aging process. Besides its link with cancer, other factors could explain the decline of CCT subunits during differentiation. In support of the disposable soma theory of aging⁶², a fascinating hypothesis is that downregulation of the TRiC/CCT complex during differentiation is part of an organismal genetic program that ensures a healthy progeny whereas somatic tissues undergo a progressive demise in their homeostasis and function. Due to the limitation of nutrients in nature, organisms divide the available metabolic resources between reproduction and maintenance of the non-reproductive soma. Evolutionary pressure has been theorized to force a re-allocation of the resources to prevent or eliminate damage to the germline and progeny, while little resources are placed on the maintenance of somatic cells⁶². In support of this hypothesis, signals from the germline modulate the aging of somatic tissues and removal of the germline extends lifespan⁴⁴. Interestingly, our results indicate that germline-lacking worms exhibit increased levels of TRiC/CCT in post-mitotic cells".

3. The first line of the abstract is a little confusing. I think that the authors mean that hESCs can replicate continuously while avoiding senescence, not - if not for senescence they would replicate continuously, which is how it reads now.

We agree with reviewer #1 and we have modified the text to make more clear

this point. The text now says: “Human embryonic stem cells (hESCs) can replicate indefinitely while maintaining their undifferentiated state and, therefore, are immortal in culture”.

Reviewer #2 (Expert in stem cell biology)

A. Summary of the key results

In this paper the authors have for the first time shown that human pluripotent stem cells have greater levels and assembly of chaperonin complex, TRiC/CCT, than differentiated cell types. These complexes are required at some level for maintaining pluripotency in human PSCs. Perturbing this complex results either in the cell death or decreased pluripotency levels. They also show some evidence of how these enhanced proteostatic capabilities of PSCs may be useful in age-related disease modeling with iPSCs, particularly for a neurodegenerative disease like Huntington's. They then complement their human PSC data using *C. elegans* model system establishing that proteostatic is also key for self-renewal potential of human ESCs as well as in aging. The advances described in this study are worthy of publication in Nature Communications after some additional data and revisions suggested below.

B. Originality and significance: if not novel, please include reference

This study is of great significance and novelty is high. The only other study that has shown that this complex has importance in ESC self-renewal potential is one that use murine embryonic stem cells and actually evaluates calreticulin-deficient mouse ESCs. Another relevant study looks at HSF1 in yeast-based therapeutic screen for Huntington's Disease and discovers TriC/CCT complex interaction with HSF1.

Faustino RS et al. Stem Cells 2010. Decoded Calreticulin-Deficient Embryonic Stem Cell Transcriptome Resolves Latent Cardiophenotype.

We apologize for not citing this relevant study in our first submission. We have now discussed these findings. The text says: “Although the exact mechanisms are still unknown, the TRiC/CCT complex could impinge upon hESC function in several (and non-exclusive) manners. TRiC/CCT assists the folding of a significant percentage of nascent proteins such as actin and tubulin^{16,45}. Thus, one possibility is that enhanced TRiC/CCT assembly is required for the proper folding of specific regulatory or structural proteins involved either in maintenance of hESC identity or generation of healthy differentiated cells. Interestingly, a study performed in calreticulin^{-/-} mouse ESCs showed a downregulation in the expression of Cct2, Cct3 and Cct7⁴⁶ in these cells. Calreticulin, a chaperone that binds to misfolded proteins preventing their export from the endoplasmic reticulum, is essential for cardiac development in mice⁴⁷ and required for proper myofibril formation during cardiomyocyte differentiation of mouse ESCs in vitro⁴⁸. Since the TRiC/CCT complex regulates folding and actin dynamics, the downregulation of Cct subunits in calreticulin^{-/-} mouse ESCs could forecast the

myofibrillar disarray observed in their cardiomyocytes counterparts⁴⁶”.

Neef DQ et al. PLoS Biol 2010. Modulation of heat shock transcription factor 1 as a therapeutic target for small molecule intervention in neurodegenerative disease.

In this study, the authors discovered a potential link between HSF1-mediated stress response and TRiC/CCT activity. The same group has recently published further insights into the mechanisms by which TRiC/CCT activity regulates HSF1 signaling (Neef et al. Cell Reports 2014). We have now discussed these two manuscripts in the context of our hsf-1 knockdown experiments. The text now says: “Notably, somatic CCT8 overexpression can partially protect this nematode from knockdown of hsf-1, a transcription factor that coordinates cellular protein-misfolding responses. The precise mechanism by which the TRiC/CCT complex ameliorates the detrimental effects triggered by loss HSF1 is not yet understood. In response to proteotoxic stress, HSF1 binds heat shock elements (HSE) in the promoters of target genes and triggers their expression⁴⁹. In mammalian cells, all the CCT subunits contain HSE and are transcriptionally activated by HSF1⁵⁰. Thus, ectopic expression of CCT8 could ameliorate the effects induced by a decrease of CCT subunits upon HSF1 knockdown. In addition, a direct regulatory interaction between TRiC/CCT activity and induction of proteotoxic-stress response by HSF1 has been recently reported^{51,52}. HSF1A, a chemical activator of HSF1, binds to the TRiC/CCT complex and inhibits its folding activity^{51,52}. Both the inactivation of TRiC/CCT complex by HSF1A or its depletion by loss of CCT subunits induce HSF1 activity⁵¹. Since TRiC/CCT chaperonin interacts with HSF1⁵¹, TRiC/CCT could have a direct repressor role in regulating HSF1⁵¹. However, a decrease in TRiC/CCT activity mediated by either HSF1A or knockdown of CCT subunits can also lead to the accumulation of misfolded proteins that trigger the HSF1-induced proteotoxic response⁵¹. In support of this hypothesis, our results suggest that increased TRiC/CCT assembly induced by CCT8 reduces the accumulation of misfolded proteins and, therefore, ameliorates the deleterious impact of reduced HSF1-mediated signaling”.

C. Data & methodology: validity of approach, quality of data, quality of presentation

The overall quality and style of presenting the data is and approach taken is valid. Experiments have been carried out with precision. The weakness in this study is related to the HD iPSC modeling portion of the paper in Figure 2 where they are trying to show an application related to the importance of knockdown of CCT subunits in Huntington's disease and accumulation of polyQ aggregates. At least CAG repeats (> 34 CAG repeats) allelic series of from 3 different HD patient donors and their iPSC cell lines are needed as true biological replicates for each group to reach statistically sound conclusions. These HD iPSC lines are now easily available from multiple repositories. The authors could also bolster their conclusions by using multiple clonal HD iPSC lines is from the CAG repeat donor. This is a norm for iPSC disease modeling field now.

Moreover, the CAG repeat length for the HD patient donor iPSC shown in this paper is not described at all.

Reviewer #2 is absolutely right, and several HD-iPSCs carrying different CAG repeat expansions are needed to reach conclusions. In our first submission, we used a control iPSC line and a HD-iPSC line (polyQ71) generated by Prof. Daley's laboratory using retroviral vectors. We have now included a new HD-iPSC line reprogrammed from the same parental fibroblast (polyQ71) using episomal vectors (HD Q71 iPSC line #2 in the manuscript). This line was obtained from NINDS Human Cell and Data Repository (NHCDR). Most importantly, we have now included three new HD-iPSCs from different patient donors (polyQ57, polyQ60 and polyQ180) and two new control iPSC lines from different donors obtained from NHCDR. These data can be found in Figure 4 and Supplementary Figures 8-10.

We apologize for not describing the CAG repeat length for the HD-iPSC line in our first submission. We have now described the CAG repeat expansion for every HD-iPSC line used in this paper (in the corresponding Figures and Methods section).

Most importantly, this paper has NOT evaluated a single phenotype or molecular mechanisms of proteotoxic stress in the relevant neural cell types from the HD iPSCs i.e. the striatal neurons that degenerate in human HD patients. The HD hESCs and hiPSCs studies published thus far have shown no phenotypes. Therefore, it is important to demonstrate relevance of disturbing the proteostasis and poly Q aggregates in a relevant neuronal cell type. Protocols for striatal neuron differentiation from iPSCs have been now published by multiple labs. PMID: 22748968; PMID: 18922775 This is necessary.

In our first submission, we did not include any data on relevant neuronal cell types from HD-iPSCs because our main focus was to study proteostasis of pluripotent stem cells. However, Reviewer #2 raises a very important point: Since neurons derived from HD-iPSCs lack HD phenotypes (i.e., neurodegeneration and accumulation of polyQ inclusions), it is interesting to examine the relevance of disturbing proteostasis in these cells.

First, we have now discussed the phenotypes (and lack of phenotypes) reported in these neuronal cells. The text now says: "HD-iPSCs represent a valuable resource to gain mechanistic insights into HD²³. Neuronal dysfunction and death occurs in many brain regions in HD, but striatal neurons expressing cAMP-regulated phosphoprotein (DARPP32) undergo the greatest neurodegeneration²⁹. HD-iPSCs can be terminally differentiated into neurons (including DARPP32-positive cells) that exhibit HD-associated phenotypes such as cumulative risk of death over time and increased vulnerability to excitotoxic stressors²³. Furthermore, proteotoxicity induced via autophagy inhibition or oxidative stress results in higher neurodegeneration of HD cells compared with controls²³. Despite these phenotypes, mutant HTT-expressing neurons present important limitations for disease modeling such as lack of robust neurodegeneration, polyQ aggregates and gene expression changes resembling HD^{23,25}. Despite the efforts

to detect polyQ aggregates under different conditions (e.g., adding cellular stressors), the presence of inclusions has not been observed in neurons derived from HD-iPSCs^{23,25}. The lack of inclusions in these cells could reflect the long period of time before aggregates accumulate in HD²³. Consistently, HD human neurons do not accumulate detectable polyQ aggregates at 12 weeks after transplantation into HD rat models whereas these inclusions are observed after 33 weeks of transplantation²⁵”.

Second, we have performed differentiation into striatal neurons following the protocol described in Ma et al (Cell Stem Cell, 2012) and induced proteasome inhibition (MG-132 treatment) as well as TRiC/CCT dysfunction (knockdown of CCTs). We differentiated successfully 3 HD-iPSCs (Q57, Q71 line #2 and Q180) and 2 control iPSCs lines (Q21 and Q33) in 4 independent experiments. We also induced differentiation of other iPSC lines but we were not able to obtain DARPP32 positive neurons from these cells in more than 2 experiments. Therefore, we have only included data from the aforementioned lines that were successfully differentiated into striatal neurons. Once differentiated into neurons, we knocked down different CCT subunits and inhibited the proteasome to examine the accumulation of aggregates. Notably, we did not observe polyQ aggregates in mutant HTT-expressing neurons either upon proteasome inhibition or knockdown of CCTs. These results are consistent with the findings reported in other studies (e.g., PMID: 22748968) where the authors could not detect polyQ inclusions even after treating the neurons with cellular stressors such as glutamate or proteasome inhibitors. As these studies suggested, the lack of inclusions in these cells could reflect the long period of time before aggregates accumulate in HD. We would like to remark that in the aforementioned manuscript (PMID: 22748968), the authors examined whether toxic stress has a higher impact on the viability of HD neurons compared with control cells. For these experiments, they used H₂O₂ (oxidative stress), proteasome inhibitors (lactacystin) and autophagy inhibitors modulating pathways involved in proteostasis. Interestingly, they observed that both oxidative stress and autophagy inhibition, but not proteasome inhibition, led to an increased amount of cell death in HD neurons when compared to control cells. Strikingly, we found that knockdown of CCTs resulted in increased cell death of HD neuronal cultures as assessed by cleaved caspase-3 quantification whereas it did not induce cell death in control neurons. In contrast, proteasome inhibition induces cell death at the same extent in both control and HD neurons. Thus, our results indicate that mutant HTT-expressing neurons are more sensitive to TRiC/CCT dysfunction and they undergo neurodegeneration before accumulating aggregates. These data has now been included in the manuscript (Supplementary Figure 11).

There is also a lack of in-lab HD-iPSC pluripotency validation data described in supplementary figures. Even though iPSCs may be procured from another lab, basic iPSC QC within the lab is required to ensure stable cytogenetics. G-band karyotype (cytogenetic stability) and STR identity analysis is required on all the PSCs used in this study including the H1 and H9 hESCs, as well as the HD-iPSCs.

We have now confirmed that all the hESC and iPSC lines used in our

experiments had a normal diploid karyotype as assessed by SNP genotyping. Please find the SNP genotyping data in “SNP genotyping” file. We think it is not necessary to include this amount of data in Supplementary Figures. If Reviewer #2 disagrees, we will include the SNP genotyping data in Supplementary Figures. We have now indicated in the Methods section that the hESCs and iPSCs used in our study have a normal karyotype. The text now says: “The hESC and iPSC lines used in our experiments had a normal diploid karyotype as indicated by single nucleotide polymorphism (SNP) genotyping”. We have also described how we performed SNP analysis in the Methods section.

STR analysis across 8 different loci indicated that the H9 and H1 hESCs used in our study matched the reported STR profile of these cells. These analyses also indicated no contamination with any other human cell lines. Please find the STR results in “STR analysis hESCs” file. We also confirmed genetic identity of all the HD-iPSCs and two control iPSCs with the corresponding parental fibroblasts by STR analysis across 8 different loci. Please find the STR results in “STR analysis HD-iPSCs” file. We have now discussed these data in the Methods Section: “Genetic identity of hESCs was assessed by short tandem repeat (STR) analysis. The H9 and H1 hESC lines used in our study matches exactly the known STR profile of these cells across the 8 STR loci analyzed. No STR polymorphisms other than those corresponding to H9 and H1 were found in the respective cell lines, indicating correct hESC identity and no contamination with any other human cell line. By STR analysis, we also confirmed correct genetic identity of the iPSCs used in our study with the corresponding parental fibroblast lines when fibroblasts were available (i.e., ND42242, ND36997, ND41656, ND36998, ND42229, ND36999 and HD Q71-iPSC line #1)”.

Unfortunately, we did not have the parental fibroblasts of control iPSC line #1 (hFIB2- iPS4) for STR profile comparison. However, we believe that the use of this line as an additional control is justified because it has similar characteristics regarding proteostasis of HTT when compared with the other two control iPSC lines. For instance, we confirmed that control iPSCs #1 only express wild-type HTT (Fig. 4a) and do not accumulate polyQ aggregates upon knockdown of CCTs (Fig. 4c). Furthermore, we also confirmed that control iPSC line #1 down-regulates the expression of CCT subunits during differentiation (Supplementary Figure 5a). As mentioned above, the control iPSCs #1 used in our experiments had a normal, diploid, male, chromosomal content as assessed by SNP analysis.

D. Appropriate use of statistics and treatment of uncertainties

This was one of the major weaknesses in this manuscript. Then n's are mentioned in each figure legend, however, it is not clearly described where the n's are derived from? Are these true biological and experimental replicates or only technical replicates? Is it independent experiments with multiple biological replicates? The statistics should be run on minimum of 3 exclusively performed independent experiments (each experiment

containing multiple biological and technical replicates).

We apologize for not clearly describing where the n's are derived from in our first submission. In the corresponding figure legends, we have now made clear that mean \pm s.e.m and statistical analyses are calculated from n independent experiments. For each figure, we performed at least three independent experiments. Each independent experiment contains multiple biological and technical replicates. In each independent experiment, biological replicates/wells were averaged for every condition. Then, the data of different independent experiments were averaged and compared across conditions/groups.

Second, it wasn't clear in the figure legends of figure 1 and 2 where and how the statistics on the data of the knock down of CCT subunits is derived from. Like, n=8, 9, and 9 for CCT2, CCT6A, CCT7. Is that 8-9 different wells within one single experiment? Or are these true independent experimental replicates of 3-5 wells, 8-9 different times? Minimum of three biological replicates would be required in 3 independent PSC lines in 1 experiment. Then the assay and phenotype data should be averaged for the group. Then each assay has to be repeated a minimum of 3 times (for every hESC/hiPSC line) in the same experiment. Then the data should be averaged and compared across the groups. Multiple wells in 1 experiment do not qualify as independent experiments.

We would like to thank Reviewer #2 for noticing this: The statistics analysis on the data of knockdown of CCT subunits was performed with n biological repeats (wells) of different independent experiments. As the reviewer indicates, this is not the right manner to analyze these data. We have now corrected this: 1) We first averaged the biological replicates from an independent experiment. 2) Then, we averaged multiple (at least three) independent experiments. 3) Finally, we compared this average across the different groups. The corrected data analysis and graphs are now shown in Figure 3 and Supplementary Figures 6-7. We have performed these series of knockdown experiments in 2 independent hESCs (H9 and H1) and 2 iPSC lines (iPSC control line #2 and HD Q71 iPSC line #1). We obtained similar results in all the lines tested. I would also like to remark that it has been extensively demonstrated that removal of individual CCT subunits affects the assembly and function of the TRiC/CCT complex. Since we found that knockdown of 4 single distinct subunits (two different shRNAs for each subunit) results in similar effects on pluripotency marker levels, our data support that enhanced TRiC/CCT complex is important for maintaining pluripotent stem cell function. In our previous manuscript version, we showed TRiC/CCT assembly experiments in two independent hESC lines (H9 and H1). Please notice that we have now added new data showing increased assembly of TRiC/CCT complex in iPSCs compared with their differentiated counterparts (Supplementary Figure 5). Therefore, we have now observed increased levels of TRiC/CCT complex in three different pluripotent stem cell lines.

For the HD iPSCs, a minimum n of 3-4 patients is required per group for comparison, which is now the standard for iPSC disease models being accepted and published in

high impact journals. The authors also need to have an equal number of control subject iPSC lines.

As mentioned above, we have now used iPSCs from 4 different patient donors (polyQ57, polyQ60, polyQ71 and polyQ180) and three different control iPSCs from different donors.

In figures 3 and 4 where *C. elegans* data is presented and the n's are described as 77/96 and so on, please clearly describe is this an aggregate (meta-analysis) of the n's over multiple experiments and same/different stocks. If so, how many?

We apologize for not making this point more clear in our first submission. We have presented our lifespan data following the standard style used in aging research manuscripts (e.g., Arantes-Oliveira, N. et al (Science, 2002), Merkwirth, C. (Cell, 2016), Lapierre, L. et al (Nat. Communications. 2013). Each graph represents a Kaplan-Meier survival plot of a single independent experiment. In each graph, experimental and control animals were grown in parallel and log-rank (Mantel-Cox) statistical test was employed to compare between populations of the experiment. The P-values refer to experimental and control animals in a single experiment. We present a representative experiment in the main text whereas the replicate experiments with their statistical analysis and also independent transgenic lines can be found in the supplementary materials.

In each experiment, we set up the assay with 96 worms for each population being tested. Considering that a portion of the population will be censored, it is advisable to start with approx. 100 worms to achieve meaningful statistics (Amrit, F.R.G. et al, Methods, 2014). From the initial worm population, the worms that are lost or burrow into the medium as well as those that exhibit 'protruding vulva' or undergo bagging were censored. $n = \text{total number of uncensored animals} / \text{total number (uncensored + censored)}$ of animals observed in each experiment.

*We have now made clear this point in the figure legend of the first figure presenting *C. elegans* data (Figure 5): "Each lifespan graph shows a Kaplan-Meier survival plot of a single representative experiment. In each graph, experimental and control animals were grown in parallel. $n = \text{total number of uncensored animals} / \text{total number (uncensored + censored)}$ of animals observed in each experiment. P-values refer to experimental and control animals in a single lifespan experiment. See **Supplementary Table 4** for statistical analysis and replicate data of lifespan experiments."*

*We have also explained this assay in further detail in the Methods section: "**Lifespan studies.** Lifespan analyses were performed as described previously¹⁰. Animals were grown at 20°C until day 1 of adulthood. 96 animals were used per condition and scored every day or every other day⁷⁸. From the initial worm population, the worms that are lost or burrow into the medium as well as those that exhibit 'protruding vulva' or undergo bagging were censored. $n = \text{total number of uncensored animals} / \text{total number (uncensored + censored)}$ of animals observed in each experiment.*

*Lifespans were conducted at either 20°C or 25°C as stated in the figure legends. For non-integrated lines DVG9, DVG41, DVG44, DVG47-50 and DVG58, GFP positive worms were selected for lifespan studies. PRISM 6 software was used for statistical analysis to determine median and percentiles. In all cases, P-values were calculated using the log-rank (Mantel–Cox) method. The P-values refer to experimental and control animals in a single experiment. In the main text, each graph shows a representative experiment. See **Supplementary Table 4** for statistical analysis and replicate data.”*

E. Conclusions: robustness, validity, reliability

Given the significant variability in differentiation of iPSC-derivatives from experiments a minimum of 3 experimental replicates are required and data averaged over those experiments.

We averaged and analyzed data of at least 3 experimental replicates in the iPSC figures presented in this manuscript.

E. Conclusions: robustness, validity, reliability

For the many specific experiments conducted the conclusions are valid. However, there is a lack of clarity regarding the significance and role of each of the subunits in human vs *C. elegans* data.

In Figure 1, in human cells it is clear the CCT8, CCT7, CCT6A and CCT2 are important for pluripotency in hESCs and assembly of TRiC/CCT complex. Here CCT1 does not seem to be critical in this process.

In Figure 2, in human cells it is clear that knockdown CCT7, CCT6A and CCT2 affects pluripotency and proteostasis. It is not clearly explained why CCT8 drops out of these experiments?

In Figure 3 and 4, in *C. elegans*, it cct-8 are the most important subunit in increasing longevity and proteotoxic stress. The human ortholog of the *C. elegans* cct-8, CCT8 does not appear to be as critical in affecting proteotoxic stress. This rationale for this should be explained.

As a result of the discrepancies in the results between the different CCT subunits across the multiple species it is very difficult to follow the paper. Importantly, the reliability of the phenotypes cannot be ascertained due to the minimal number of PSCs utilized in each experiment group.

*Reviewer #2 is right and our previous data could lead to think that there are discrepancies in the results between the different CCT subunits across species. We have now included new experiments and re-written the manuscript to make clear the role of CCT8 as an activator of TRiC/CCT assembly in both human and *C. elegans* cells.*

Given that knockdown of a single subunit is sufficient to induce TRiC/CCT function, we thought that knockdown of three distinct CCT subunits (CCT2, CCT6A and CCT7) was sufficient to demonstrate that increased TRiC/CCT assembly modulates

pluripotency and proteostasis in human PSCs. Since we found that CCT8 is sufficient to increase TRiC/CCT assembly and extend lifespan in C. elegans, we agree with Reviewer #2 that it is important to show the effects of knockdown of CCT8 in human PSCs to improve the clarity of the paper. We have now included CCT8 knockdown experiments to examine pluripotency marker levels in hESCs and iPSC lines (Figure 3a and Supplementary Figures 6b-c). As with CCT2, CCT6A or CCT7, knockdown of CCT8 affects pluripotency to a similar extent. We have also examined the impact of CCT8 knockdown on polyQ aggregation in HD-iPSCs (Figure 4d). Consistent with the hypothesis that increased TRiC/CCT assembly modulates proteostasis of iPSCs, loss of CCT8 induces a similar polyQ aggregation compared with CCT2, CCT6A or CCT7 knockdown. As discussed above, we have now examined the effects of CCT knockdown in additional pluripotent stem cell lines and obtained similar results (2 independent hESCs (H9 and H1) and 2 iPSC lines (iPSC control line #2 and HD Q71 iPSC line #1). In addition, we have also added several HD and control iPSCs to study the effects of TRiC/CCT dysfunction on mutant HTT aggregation. We believe that these experiments support our conclusion that high levels of TRiC/CCT are important for pluripotency and proteostasis of PSCs.

*Although our data indicates that the downregulation of other CCT subunits could also modulate the decline in TRiC/CCT assembly during differentiation and aging (as we discussed in the main text), our results clearly emphasize the positive role of CCT8 as a key activator of TRiC/CCT assembly. In this regard, we have performed a comparison between the levels of the different CCT subunits in hESCs and their differentiated counterparts (**Fig. 2b**). This comparison indicates that CCT8 could be an activator of TRiC/CCT assembly rather than a structural limiting factor. Thus, this analysis in human cells is consistent with our findings in C. elegans, where CCT8 overexpression has the strongest impact on lifespan compared with other subunits. Most importantly, we have now included native gel data showing that CCT8 is sufficient to trigger TRiC/CCT assembly in both mammalian cells (**Fig. 2c**) and somatic tissues of C. elegans (**Fig. 5e**). We believe that these experiments strengthened the role of CCT8 as a potent activator of TRiC/CCT assembly and proteostasis. Hence, it is now more clear why this specific subunit is an interesting target to be overexpressed in somatic tissues with the aim to increase proteostasis and examine its effects in the context of aging/proteotoxic resistance of post-mitotic tissues. We have now added the aforementioned results and re-written the manuscript to strengthen the link between CCT8, TRiC/CCT assembly and increased proteostasis across species.*

Suggested improvements: experiments, data for possible revision

1. Figure legend 2C. Knock down of CCT6A and CCT7 does not appear to induce expression of endoderm markers at all. Rather it should say ectoderm. Please correct this. Also, it is rather curious as to why the knockdown does not affect mesoderm and endoderm expression in 2C. The authors need to explain clearly the rationale behind

this?

We apologize for this mistake. We have followed Reviewer's suggestion and included a larger set of germ layer specific genes (please see suggestion #7). After performing these experiments, we observed an increase in specific markers of the three germ layers upon knockdown of CCT subunits. Thus, we cannot conclude that loss of CCT subunits induces the expression of markers of a specific germ layer. We have now discussed these results in the text: "In addition, loss of CCT levels induced the expression of markers of the distinct germ layers (Fig. 3c and Supplementary Fig. 7). Since we observed an up-regulation in specific markers of the three germ layers (Fig. 3c and Supplementary Fig. 7), our data suggest that hESCs undergo a decline of pluripotency upon knockdown of CCT subunits but they do not differentiate into a particular cell lineage".

2. Clarify the statistics as stated previously.

As discussed above, we have now clarified the statistics in each figure legend.

3. Evaluate and include publicly available transcriptome data regarding levels and relevance of the chaperome. There is also GEO data from published HD iPSC studies that have shown some or no phenotypes in the differentiated neurons.

A) We have now discussed in more detail the gene expression analysis reported in Brehme et al (Cell Reports, 2014) regarding changes in the chaperome network during human brain aging. We have also included more information regarding the relevance of these changes in the context of proteostasis-related neurodegenerative disorders. The introduction now says: "In this study, we focused on the chaperome network, a key node of proteostasis. The human chaperome is formed by 332 chaperones and co-chaperones that regulate the folding and function of proteins¹³. The binding of chaperones to nascent proteins assists their folding into the correct structure. Furthermore, chaperones assure the proper folding and cellular localization of proteins throughout their life cycle¹⁴. Gene expression analysis of human brain aging shows a striking repression of 32% of the chaperome, including ATP-dependent chaperone machines such as cytosolic HSP90, HSP70 family members (e.g., HSPA8, HSPA14) and subunits of the TRiC/CCT complex. In contrast, 19.5% of the chaperome is induced during human brain aging¹³. In addition, these repression and induction are enhanced in the brains of those with HD, Alzheimer's or Parkinson's disease compared with their age-matched controls¹³. Notably, knockdown of specific chaperome components that are repressed during aging such as CCT subunits, HSPA14 or HSPA8 induces proteotoxicity in HD C. elegans and mammalian cell models¹³. Therefore, defining differences in the levels and regulation of chaperone machines between immortal hESCs and their differentiated counterparts could be of central importance not only for understanding hESC identity but also the aging process". (...) "Since the levels of CCT subunits are further decreased in somatic tissues during organismal aging¹³, we examined whether

modulation of CCT8 can delay the aging process and proteostasis dysfunction by using C. elegans as a model organism”.

*B) We have now included and evaluated our transcriptome data comparing hESCs with NPCs and neurons at different weeks of differentiation. The text now says: “To examine changes in the chaperome network during differentiation, we performed quantitative analysis of both the transcriptome and proteome comparing hESCs with their neural progenitor cell (NPC) and neuronal counterparts (**Supplementary Fig. 1-2 and Supplementary Tables 1-2**). In our transcriptome analysis, we identified 279 chaperome components. Among them, 119 genes were down-regulated and 44 genes were up-regulated during differentiation into NPCs (**Supplementary Fig. 1 and Supplementary Table 1**).”*

C) We have now discussed the phenotypes (and lack of phenotypes) in HD neurons reported in the HD iPSC Consortium’s manuscript (PMID: 22748968), including the gene expression analysis comparing HD neurons with controls. The text now says: “HD-iPSCs represent a valuable resource to gain mechanistic insights into HD²³. Neuronal dysfunction and death occurs in many brain regions in HD, but striatal neurons expressing cAMP-regulated phosphoprotein (DARPP32) undergo the greatest neurodegeneration²⁹. HD-iPSCs can be terminally differentiated into neurons (including DARPP32-positive cells) that exhibit HD-associated phenotypes such as cumulative risk of death over time and increased vulnerability to excitotoxic stressors²³. Furthermore, proteotoxicity induced via autophagy inhibition or oxidative stress results in higher neurodegeneration of HD cells compared with controls²³. Despite these phenotypes, mutant HTT-expressing neurons present important limitations for disease modeling such as lack of robust neurodegeneration, polyQ aggregates and gene expression changes resembling HD^{23,25}”.

4. Incorporate allelic series of CAG repeat HD iPSC lines.

As mentioned above, we have now described the CAG repeat expansion for every HD-iPSC line used in this paper (in the corresponding Figures and Methods section).

5. Clarify differences in species and different CCT subunits and their relevance to proteotoxic stress.

As discussed previously, we have re-written the manuscript and added new data to clarify this. We have now made clear that by studying proteostasis of hESCs we defined CCT8 as a potential candidate to increase the assembly of the TRiC/CCT complex. The overexpression of this single subunit triggers TRiC/CCT assembly in C. elegans somatic tissues and has beneficial effects at the organismal level (e.g., lifespan extension, delay of age-associated decline in proteostasis and amelioration of polyQ-expanded associated phenotypes). Although our data indicates that the downregulation of other CCT subunits could also modulate the decline in TRiC/CCT assembly during

differentiation and aging, here we focused on the positive impact of increased levels of CCT8 in TRiC/CCT assembly, proteostasis and lifespan.

6. It would be ideal if the authors can include levels CCT subunit proteins of WT and HD BAC mice primary striatal neurons.

We agree with Reviewer #2 that these experiments could generate interesting data. Although we started the paperwork to obtain the license to bring these mouse models to our animal facility and perform these experiments, this procedure will take several months and we would significantly miss the deadline to submit the revision for this manuscript. We tried to solve this issue by contacting different colleagues in the area but they only have the R6/2 mouse strain, which is not the ideal model to study HD. However, a recent paper by Langfelder et al (Nature Neuroscience, 2016) has showed an integrated genomics and proteomics study of knock-in HD mouse models expressing the human HTT exon1 carrying different CAG lengths. In this paper, the authors found an up-regulation in CCT8 at both mRNA and protein levels in the striatum of polyQ175 HD knock-in mice. In contrast, the levels of other subunits were not changed compared with control strains. We have now discussed these findings in the Discussion section. The text says: "Recently, a link between CCT8 and mutant HTT expression was observed in an integrated genomics and proteomics study of knock-in HD mouse models expressing the human HTT exon1 carrying different CAG lengths (polyQ20, Q80, Q92, Q111, Q140 or Q175)⁵³. Notably, highly expanded-polyQ HTT exon1 (Q175) induces a significant increase of CCT8 at both transcript and protein levels in the striatum of young mice (6-months) whereas other subunits were not altered⁵³. In contrast, CCT8 induction was not observed in knock-in mice expressing mutant HTT with lower than 175 polyQ repeats. These findings suggest that up-regulation of CCT8 could be a compensatory mechanism to protect from polyQ aggregation".

7. Explain why CCT6A and CCT7 knockdown is mostly ineffective in mesodermal and endodermal gene expression. Perhaps include a larger set of germ layer specific genes.

We appreciate this suggestion. With the limited set of germ layer markers analyzed in our first submission, our results suggested that knockdown of CCTs induces an increase in ectodermal markers. However, we have now included a larger set of germ layer specific genes and found that specific markers of the other germ layers are also increased. Thus, it appears that hESCs do not differentiate into a specific cell lineage upon knockdown of CCT subunits.

8. Show other relevant controls of inducing proteotoxic stress in human ESC/iPSCs and differentiated neurons with inhibition of proteasomes (bortezomib) or inhibition of Hsp90 (geldanamycin).

Since we had previously established the working concentration for MG-132 treatment in the laboratory, we used this proteasome inhibitor instead of bortezomib. In Supplementary Figure 8, we have now shown that MG-132 treatment induces the accumulation of polyQ aggregates in HD-iPSCs by both filter trap and immunohistochemistry analysis. We would like to remark that a previous study (reference 25, Jeon et al, Stem Cells, 2012) already showed that MG-132 treatment triggers polyQ aggregation in HD-iPSCs by immunohistochemistry experiments. This publication has been appropriately cited in our manuscript. We also treated HD neurons with proteasome inhibitors (Supplementary Figure 11). As observed in The HD iPSC Consortium's manuscript (Cell Stem Cell, 2012), we were not able to detect polyQ aggregates in HD neurons even upon the treatment with cellular stressors. Moreover, we found that proteasome inhibition induces cell death in both control and HD neurons. These experiments were performed in multiple HD and control neurons in 4 independent differentiation experiments.

Regarding HSP90 inhibitors (geldanamycin and its derivatives such as 17-AAG), several studies showed that these reagents induce a heat-shock response that eventually reduces polyQ aggregation in HD mammalian cells and *D. melanogaster* models (e.g., Herbst et al, Neurodegenerative Dis. 2007). Here we used 17-AAG (a pharmacologically improved geldanamycin derivative) because this compound has been already tested in ESCs to study the impact of Hsp90 on pluripotency (Bradley et al, Stem Cells 2012). We have used the same concentration (250 nM 17-AAG for 48h) as published in the aforementioned paper. As expected, we were not able to detect changes in polyQ aggregation upon 17-AAG treatment (please see figure below) because HD-iPSCs do not accumulate polyQ aggregates in normal conditions. Likewise, we did not observe changes in HD neurons upon 17-AAG treatment. We have not included these data because we think this information is not relevant for our manuscript. If Reviewer #2 believes it is necessary, we can include the following text and figure in our manuscript:

"In HD organismal models and mammalian cells, the inhibition of HSP90 induces heat-shock response and reduces polyQ-expanded protein aggregation (Herbst et al, Neurodegenerative Dis. 2007). We found that the treatment with an inhibitor of HSP90 chaperones (Bradley et al, Stem Cells 2012) did not further decrease the signal observed in HD-iPSCs by filter trap, reinforcing that these cells do not accumulate detectable aggregates of mutant HTT (Supplementary Figure 17-AAG treatment)."

SDS-resistant polyQ aggregates

iPSCs treated with 17-AAG. Filter trap analysis shows that the treatment of HD-iPSCs with 17- N-Allylamino-17-demethoxygeldanamycin (17-AAG), a HSP90 inhibitor, does not affect polyQ aggregation (detected by anti-polyQ-expansion diseases marker antibody). Cells were treated either with 250 nM 17-AAG for 48 h or 5 μ M MG-132 (proteasome inhibitor) for 16 h.

F. References: appropriate credit to previous work?

References have been appropriately cited except for some of the citations mentioned in this review.

As explained in section B, we have now discussed this relevant work in our manuscript.

G. Clarity and context: lucidity of abstract/summary, appropriateness of abstract, introduction and conclusions

Overall, the paper is apprehensible and well-written including the abstract. Introduction is sound, although a more clear description of the difference and relevance of the CCT subunits in engendering proteotoxic stress across human cells and *C. elegans* should be clarified better. The conclusions and their interpretations about many of in vitro results with HD iPSCs are much overstated without additional HD iPSC donor lines.

We have addressed these important points. The manuscript is now more clear and the conclusions are supported with additional controls.

Reviewer #3 (Expert in *C. elegans* aging)

The observations that TRiC/CCT complexes are important for hESC pluripotency and can mildly increase lifespan when overexpressed in *C. elegans* are interesting, and it is always nice to see the combination of mammalian and invertebrate mechanistic experiments in one study. However, there is some convoluted logic that enters the paper with the introduction of *C. elegans* to the story (see 2-5 below):

1. Perhaps I missed it, but do TRiC/CCT components decline with passage, increasing senescence, and loss of pluripotency (not just differentiation, as shown)? This would be an important indication of the ability of TRiC/CCT components to maintain "stemness".

We agree with Reviewer #3 that these are very important points to be clarified. We have now re-written the manuscript to make clear that TRiC/CCT components decrease during differentiation and that hESCs maintain their high expression of CCT subunits with passage while they are in an undifferentiated/pluripotent state.

Self-renewal: We have now examined whether the expression of CCT subunits decrease in hESCs with passage (Figure 1g, h). We found that when hESCs are maintained in their pluripotent state, the expression of CCTs does not decline with passage. The text now says: “Consistent with the ability of hESCs to self-renew indefinitely while maintaining their undifferentiated state^{6,7}, the expression of CCT subunits and pluripotency markers did not decline with passage (Fig. 1g, 1h). Taken together, our results indicate that hESCs are able to maintain enhanced expression of CCT subunits under unlimited proliferation in their undifferentiated state. However, the levels of subunits such as CCT8 or CCT2 decrease when hESCs differentiate into distinct cell lineages. Thus, increased levels of CCT subunits could be an intrinsic characteristic of hESCs linked to their immortality and identity”.

Increasing senescence: Pluripotent stem cells can replicate indefinitely while maintaining their pluripotency. Otherwise, they die or differentiate. Accordingly, loss of proteostasis induces cell death or differentiation. However, pluripotent stem cells cannot undergo senescence (Miura et al, Aging Cell 2004). Therefore, senescence (defined as growth arrest) cannot be induced in pluripotent stem cells. In our previous version, we mentioned the ability of hESCs to avoid senescence in the abstract and introduction. Since this can lead to a lack of clarity, we have now removed any mention to senescence. [REDACTED]

Loss of pluripotency: Pluripotency, by definition it is lost when hESCs differentiate. Typically, measured by decline in gold standard markers of pluripotency such as OCT4 and NANOG that occurs when hESCs differentiate. Therefore, hESCs cannot lose pluripotency without differentiation or cell death. When we performed our experiments at different passages, we kept hESCs in their undifferentiated/pluripotent state by removing colonies showing differentiation (assessed by changes in morphology). The observation that hESCs maintain high levels of both pluripotency markers and CCT subunits after 50 passages (Figure 1g, h), indicates a link between pluripotency status and high expression of CCTs. In other words, our results suggest that hESC express high levels of CCT subunits in their pluripotent/undifferentiated state but the expression of CCTs is downregulated during differentiation.

Taken together, we believe that the link between pluripotency, differentiation and high levels of CCT subunits is now more clear after re-writing the manuscript and including new data regarding the levels of CCT subunits and pluripotency markers at different passages.

2. The observation that germline-less worms, particularly *glp-1* mutants, have lower levels of CCT components is odd and undermines the authors' premise, as *glp-1* worms are long-lived, yet these data suggest that CCT components are dispensable for this long lifespan. Please explain.

*We agree with Reviewer #3 that the levels of CCTs in whole-animal *glp-1* worms could suggest that CCT components are dispensable for the long lifespan of these mutant worms. We have now assessed whether CCT components regulate the lifespan of *glp-1* mutants (Fig. 7b). As with *daf-2* and *eat-2* mutants, we found that loss of function of *TRiC/CCT* reduces the lifespan of germline-lacking *glp-1* worms (Fig. 7b). We have now included these data in the manuscript (Fig. 7b). As reviewer #3 indicates, these results raise an intriguing question: How can we explain that *glp-1* mutants are long-lived and *TRiC/CCT* is required for this phenotype when whole *glp-1* animals express lower amounts of CCT subunits compared with wild-type worms? Our immunohistochemistry analysis of CCT expression suggests a possible explanation for this paradox. In wild-type worms, we have found increased levels of CCT subunits in the germline when compared to the intestine (Supplementary Fig. 12a). Notably, the levels of CCTs in the intestine of *glp-1* germline-lacking mutants are increased compared with the intestine of wild-type worms (Fig. 7c, d). In addition, we also observed increased expression of *cct-8* in somatic tissues of *glp-1* mutants as assessed by transcriptional reporter experiments (Fig. 7d). Thus, we hypothesize that lack of germline in *glp-1* mutants could result in a somatic increased expression of CCT subunits required for long lifespan. However, this increased somatic expression cannot be detected when whole animals are analyzed. Given the high expression of CCT subunits in germ cells (a tissue that represents a high percentage of the organism in wild-type worms), a possible explanation is that somatic up-regulation of CCTs is not sufficient to compensate the decrease in CCT levels observed in whole animals when the germline is removed. As Reviewer #3 explains in the next comment, conclusions about CCT expression levels in germline compared with somatic tissues cannot be drawn from whole-organism experiments. For this reason, we have removed the whole-animal analysis comparing germline-lacking worms with controls. Instead, we included the aforementioned immunohistochemistry and transcriptional reporter experiments.*

3. Furthermore, the lower level of *cct* subunits in germline-less animals does NOT suggest that the *cct* subunits are expressed in germ cells--there is an immense amount of signaling between germline and somatic cells, so *cct* subunits could be expressed in somatic cells and regulated by germline proliferation status. No such conclusion can be drawn about location of CCT subunit expression from whole-animals experiments shown in Fig. S6. This line of reasoning must be deleted, and furthermore, the location of *cct*

expression should be identified.

*Reviewer #3 is absolutely right. We apologize for concluding that CCT subunits are expressed in the germline just by analyzing the levels of CCT subunits in extracts from whole animals. Indeed, growing evidence indicates that signals from the germline regulate the expression of specific proteins in somatic tissues. We have deleted this line of reasoning, removed the whole-animal qPCR experiments and performed new experiments to study the location of CCT expression. I would like to remark that previous studies reported that CCT transcripts are detected in most tissues (including the germline) and developmental stages (Hill et al, Science (2000)). We have now performed immunohistochemistry experiments to assess the expression of CCT components in the germline of adult worms. For this purpose, we used an antibody to CCT1 subunit and found that CCT1 is highly expressed in the germline of wild-type worms compared with the intestine (**Supplementary Fig. 12a**). This antibody has been profiled for use in *C. elegans* by Abcam. We validated by western blot analysis that anti-CCT1 recognizes this subunit in worms at the correct molecular weight. Furthermore, native gel experiments confirmed that this antibody recognizes CCT1 in its monomeric form as well as part of the TRiC/CCT complex.*

*To assess the expression of CCT subunits in somatic tissues of adult *C. elegans*, we have now generated a GFP transcriptional reporter construct for *cct-8* gene. In this regard, we would like to remark that the somatic expression of other CCT subunits (e.g., CCT1, CCT2, CCT7) has been previously characterized using GFP transcriptional reporters in other studies (Lundin et al, Developmental Biology (2008) and McKay et al, Cold Spring Harbor symposia on quantitative biology (2003)). As observed in these publications, we did not find GFP expression in germ cells as a result of germline silencing of transgenes. Our reporter experiments confirmed wide expression of *cct-8* in somatic cells including neurons or body wall muscle cells. We have now included these data in **Supplementary Fig. 14**. We have also made clear that somatic expression pattern of *cct-8* resembles other *cct* subunits. The text now says: "To assess the expression of CCT subunits in somatic tissues of adult *C. elegans*, we generated a GFP transcriptional reporter construct for *cct-8* gene. Although we did not observe GFP expression in germ cells as a result of germline silencing of transgenes^{37,38}, these experiments confirmed wide expression of *cct-8* in somatic cells including neurons or body wall muscle cells (**Supplementary Fig. 14**). Somatic expression pattern of *cct-8* resembled other *cct* subunits (e.g., *cct-1*, *cct-2* and *cct-7*) showing a high expression in pharynx and tail^{32,38}(**Supplementary Fig. 14**)."*

4. The connection between hESC and *C. elegans* is further strained by the fact that the authors suggest that TRiC/CCT is required for "stemness" but the tissues assayed in *C. elegans*--that is, those that determine longevity--are all differentiated and post-mitotic.

*We agree with Reviewer #3 that the connection between our hESC and *C. elegans* experiments was not clear in our first version of the manuscript. We apologize for this and we have now re-written the manuscript to make clear this link. In summary,*

the reason why we focused on C. elegans somatic tissues is because we wanted to mimic proteostasis of stem cells in post-mitotic tissues with the aim to define novel mechanisms of lifespan extension and proteotoxic resistance. This is one of the novel and most important aspects of our manuscript.

The 3 main hypotheses of our paper are:

- hESCs have increased proteostasis.*
- Therefore, hESCs could represent a novel model to identify mechanisms that promote proteostasis.*
- Mimicking these mechanisms in somatic tissues can increase proteostasis of post-mitotic cells and delay the aging of somatic tissues.*

Indeed, our results indicate that hESCs have increased assembly of the TRiC/CCT complex but this assembly is downregulated during differentiation. By studying proteostasis of hESCs, we found that modulation of CCT8 is a powerful mechanism to increase TRiC/CCT assembly in mammalian cells. We have now added these results (Fig. 2b-c), highlighting CCT8 as an interesting candidate to promote proteostasis. Since proteostasis is downregulated during differentiation and further decreases during aging, we hypothesized that mimicking proteostasis of hESCs in somatic tissues by inducing TRiC/CCT could delay the aging of post-mitotic cells. In this regard, we have now added new experiments showing that CCT8 overexpression also triggers TRiC/CCT assembly in C. elegans somatic tissues (Fig. 5e), indicating that this regulatory mechanism is evolutionary conserved. Our results suggest that increased TRiC/CCT assembly in somatic tissues extends lifespan particularly under proteotoxic conditions. Moreover, increased levels of TRiC/CCT complex (via CCT8 overexpression) reduce polyQ aggregation in the neurons of C. elegans. Taken together, we believe that the link between our work on hESCs and C. elegans somatic tissues is now clear after re-writing the manuscript and strengthened with our new experiments (e.g., analysis of TRiC/CCT assembly upon CCT8 overexpression in both hESCs and C. elegans, etc).

5. What, if any, is the role of TRiC/CCT in the mitotic germline of C. elegans? This would seem to be the more natural correlate of hESCs.

Indeed, studying the impact of TRiC/CCT in the mitotic germline of C. elegans would seem to be the more natural correlate of hESCs. However, we used C. elegans as an organismal model with a completely different goal. As discussed above, our aim was to assess whether mimicking the TRiC/CCT complex of hESCs into somatic tissues can promote proteostasis in post-mitotic cells and extend longevity. This is the innovative aspect of our manuscript.

The role of CCT in embryogenesis, development and developing gonad has been extensively studied in C. elegans (e.g., Lundin et al. Developmental Biology 2007). These studies showed that TRiC/CCT complex is required for embryogenesis and larval development. In C. elegans, loss of CCT subunits causes a variety of defects in cell division (Lundin et al. Developmental Biology 2007). For instance, it affects pronuclear

rotation spindle positioning, meiosis, chromosome aggregation and cytoplasmic streaming. TRiC/CCT dysfunction causes severe defects in developing gonad and results in sterility. These effects are partially mediated by a collapse in microtubule function as a consequence of diminished folding of tubulin by TRiC/CCT³¹. Interestingly, loss of CCT subunits causes more severe phenotypes than knockdown of prefoldin, another chaperone required for the proper folding of tubulin. These findings suggest that TRiC/CCT mediates embryogenesis and development not only by assuring the proper folding of tubulin but also other proteins, consistent with a broader role of this complex in proteostasis. We think that this evidence supports the important role of the TRiC/CCT complex in hESCs. Therefore, we have now discussed these papers in our manuscript. The text now says: "The role of TRiC/CCT complex in proliferating cells during *C. elegans* development has been extensively studied³². Disruption of TRiC/CCT assembly by knockdown of different CCT subunits causes a variety of defects in cell division and results in embryonic lethality³²⁻³⁵. These effects are partially mediated by a collapse in microtubule function as a consequence of diminished folding of tubulin by TRiC/CCT³². In addition, loss of different cct subunits during post-embryonic developmental stages results in larval arrest, body morphology alterations as well as defects in developing gonads and sterility, indicating a key role of the TRiC/CCT complex in *C. elegans* development³²".

Although our main goal is to study whether increasing TRiC/CCT assembly in somatic tissues mimics the proteostasis of hESCs, we agree with Reviewer #3 that it is important to define the expression of CCT subunits in the germline of adult worms compared with somatic tissues. Likewise, it is of interest to examine whether CCTs are required for *C. elegans* adult germline. We have now performed these experiments and obtained results that link proteostasis of hESCs and *C. elegans* germ cells. The text now says: "In adult worms, the only proliferating cells are found in the germline whereas somatic tissues are formed exclusively by post-mitotic cells³⁶. To examine the expression of CCTs in germ cells, we dissected the germline and intestine from adult *C. elegans* and compared the levels of CCT subunits by immunohistochemistry. We found that CCT-1 subunit is enhanced in germ cells compared with the intestine (**Supplementary Fig. 12a**), suggesting that CCTs are highly expressed in the germline. Notably, knockdown of cct subunits during adulthood dramatically decreased the number of germ cells destabilizing the germline (**Supplementary Fig. 12b**). Accordingly, we observed a dramatic decrease in the number of laid eggs after 2 days of cct RNAi treatment during adulthood (data not shown). Overall, these data suggest that high levels of TRiC/CCT complex are essential for proliferating cells and germline stability".

Yours sincerely,

David Vilchez

Reviewer #1 (Remarks to the Author)

This is a substantially revised and greatly improved version of the manuscript, which includes important new data and helps address some previously unaddressed questions.

I think this paper will significantly contribute to the ongoing discussion regarding the role of proteostasis in stem cell fitness and differentiation.

Reviewer #2 (Remarks to the Author)

Somatic increase of CCT8 mimics proteostasis of embryonic stem cells and extends organismal longevity in *C. elegans*

Revision 1 comments:

The authors have done excellent work bolstering their manuscript with new experiments and addressing the reviewer critiques. The inclusion of greater number and allelic series of HD iPSC lines is a positive. As a result I recommend the manuscript for publication in Nature Communications after very minor revisions suggested below.

I suggest changing in title to "human pluripotent stem cells" from "embryonic stem cells" highlight the fact that this process was demonstrated in human cells and replicated in *C. elegans*.

I would suggest moving the proposed Supp Figure 17 from the reviewer responses to the supplementary file.

The authors have done an excellent work in responding to the reviewer critiques. I would suggest moving the proposed Supp Figure 17 from the reviewer responses to the supplementary article file.

Reviewer #3 (Remarks to the Author)

The authors addressed all of my concerns, and I appreciate the addition of new data, particularly on the *glp-1* animals and tissue-specific (rather than whole-animal) levels of the CCT complex.

Response to Reviewers:

Reviewer #1 (Remarks to the Author):

This is a substantially revised and greatly improved version of the manuscript, which includes important new data and helps address some previously unaddressed questions. I think this paper will significantly contribute to the ongoing discussion regarding the role of proteostasis in stem cell fitness and differentiation.

Reviewer #2 (Remarks to the Author):

Somatic increase of CCT8 mimics proteostasis of embryonic stem cells and extends organismal longevity in *C. elegans*

Revision 1 comments:

The authors have done excellent work bolstering their manuscript with new experiments and addressing the reviewer critiques. The inclusion of greater number and allelic series of HD iPSC lines is a positive. As a result I recommend the manuscript for publication in Nature Communications after very minor revisions suggested below.

I suggest changing in title to "human pluripotent stem cells" from "embryonic stem cells" highlight the fact that this process was demonstrated in human cells and replicated in *C. elegans*.

We have now changed the title to “Somatic increase of CCT8 mimics proteostasis of human pluripotent stem cells and extends C. elegans lifespan”.

I would suggest moving the proposed Supp Figure 17 from the reviewer responses to the supplementary file.

*We have now moved this figure from the reviewer responses to Supplementary Information (**Supplementary Fig. 8a**)*

The authors have done an excellent work in responding to the reviewer critiques. I would suggest moving the proposed Supp Figure 17 from the reviewer responses to the supplementary article file.

Reviewer #3 (Remarks to the Author):

The authors addressed all of my concerns, and I appreciate the addition of new data, particularly on the glp-1 animals and tissue-specific (rather than whole-animal) levels of the CCT complex.